# DISCO: learning to DISCover an evolution Operator
# for multi-physics-agnostic prediction

Rudy Morel [1]   Jiequn Han [1]   Edouard Oyallon [1 2]

## Abstract

We address the problem of predicting the next states of a dynamical system governed by *unknown* temporal partial differential equations (PDEs) using only a short trajectory. While standard transformers provide a natural black-box solution to this task, the presence of a well-structured evolution operator in the data suggests a more tailored and efficient approach. Specifically, when the PDE is fully known, classical numerical solvers can evolve the state accurately with only a few parameters. Building on this observation, we introduce DISCO, a model that uses a large hypernetwork to process a short trajectory and generate the parameters of a much smaller operator network, which then predicts the next states through time integration. Our framework decouples dynamics estimation — i.e., DISCovering an evolution Operator from a short trajectory — from state prediction — i.e., evolving this operator. Experiments show that pretraining our model on diverse physics datasets achieves state-of-the-art performance while requiring significantly fewer epochs. Moreover, it generalizes well to unseen initial conditions and remains competitive when fine-tuned on downstream tasks. The code will be made publicly available upon publication at https://github.com/RudyMorel/DISCO.

## 1. Introduction

Modeling dynamical systems from data is a highly active area of research with the potential to significantly reduce computational costs (Kidger, 2021) and limit the need for ad-hoc engineering to predict future states (Farlow, 2012).

In this paper, we consider dynamical systems described by Partial Differential Equations (PDEs). The conventional data-driven approach typically involves "learning" a fixed dynamic model from a large number of trajectories obtained from various initializations, which is often referred to as Neural Operator learning (Kovachki et al., 2023; Boullé & Townsend, 2023). However, in many practical scenarios, the exact governing physical laws are unknown and may vary across different trajectories, making such methods unsuitable and requiring more flexibility.

To address this limitation, we study the problem of predicting the next states of a PDE-governed dynamical system from a few successive past state observations, where the underlying PDE is unknown and can vary across different trajectories. This task called *multi-physics-agnostic prediction* has recently gained attention with the effort to build foundational models for PDEs (McCabe et al., 2024; Herde et al., 2024). It requires both estimating the underlying dynamic (inverse problem) and integrating it over time (forward problem), which are typically treated independently in existing literature (Hsieh et al., 2019; Blanke & Lelarge, 2023). Tackling multi-physics-agnostic prediction requires facing at least three types of *variability* in the data:

**(1)** The PDE class for the dynamics (e.g., Navier-Stokes),

**(2)** The coefficients of the PDE (e.g., Reynolds number),

**(3)** The initial conditions of the trajectory.

A natural way to handle these three levels of variability is to use models that process a context — i.e., a sequence of preceding states — to infer the underlying dynamics. Transformers, with their strong sequential learning capabilities (Vaswani et al., 2017), excel at this through In-Context Learning (Brown et al., 2020), allowing them to adapt to different dynamics using prior state information. This adaptability has enabled transformers to achieve remarkable performance in natural language processing, where context plays a crucial role (Dubey et al., 2024). Recent works (Yang et al., 2023; Liu et al., 2023; Yang & Osher, 2024; McCabe et al., 2024; Serrano et al., 2024) have also shown that transformers, when trained on diverse PDEs and initial conditions, can predict future states across different contexts, establishing them as powerful tools for data-driven

---

[1]Center for Computational Mathematics, Flatiron Institute, New York [2]Sorbonne Université, CNRS, ISIR, Paris- France. Correspondence to: Rudy Morel <rmorel@flatironinstitute.org>.

*Proceedings of the 42$^{nd}$ International Conference on Machine Learning*, Vancouver, Canada. PMLR 267, 2025. Copyright 2025 by the author(s).

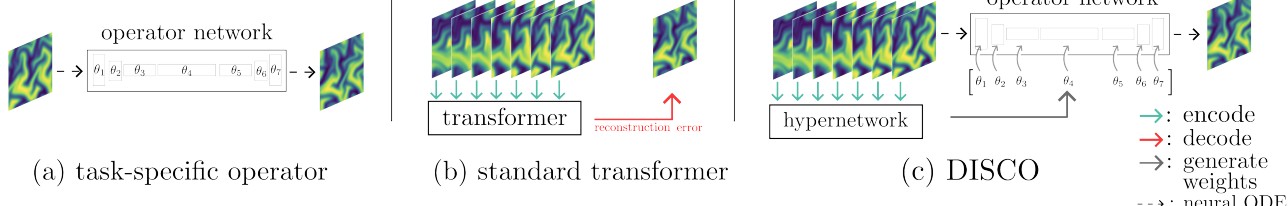

*Figure 1.* (a) A task-specific learned operator must be re-trained on unseen Physics. (b) A transformer (e.g., MPP) must learn both an encoder and a decoder, adding reconstruction error. (c) DISCO infers operator parameters without relying on an auto-encoder.

modeling of dynamical systems in physical domains.

However, applying standard transformers to physical systems is challenging. In order to process a context of potentially high-resolution frames, they typically require to reduce spatial resolution through an encoder, which must then be inverted by a decoder. Learning the decoder on physics data remains a highly non-trivial task (Olmo et al., 2022). As a result, transformers often require vast amounts of data to avoid overfitting and can struggle with predicting out-of-distribution trajectories, leading to instabilities over multi-step rollouts (McCabe et al., 2023; 2024). In contrast, classical solvers excel when the governing dynamics are known, evolving the system accurately with minimal parameters and without any data training. Such methods often preserve key structural properties of physics, such as continuous-time evolution and translation equivariance — as described in (Mallat, 1999) — where a spatial translation of the initial condition results in the same translation of the solution in the absence of boundary conditions. In contrast, transformers do not naturally inherit these properties. This lack of structure preservation contributes to transformers' limitations in physical applications, particularly in terms of training efficiency and stability.

In this work, we propose a novel model named DISCO that combines the best of both worlds by learning to DISCover an evolution Operator from a context of successive states of a trajectory. Specifically, a large transformer hypernetwork processes a data context to generate parameters for a much lower-dimensional neural-PDE-like solver (Chen et al., 2018), which then predicts the next states through time integration. In this manner, we decouple context dynamics estimation from state prediction, unlike previous transformer-based methods where both are entangled. Our approach can be viewed as a meta-learning algorithm (Thrun & Pratt, 1998; Finn et al., 2017; Koupaï et al., 2024), designed to self-adapt to specific contexts, or tasks. However, unlike recent meta-learning methods, which faces only variability **(2)** and **(3)** (Koupaï et al., 2024), our framework inherently handles all levels of variability **(1)**, **(2)**, and **(3)**. Moreover, prior meta-learning approaches have not been scaled to high-resolution data, where directly learning an operator remains challenging. The operator network used in

the solver respects the underlying continuous-time nature of the physics and preserves spatial translation equivariance through the use of a U-Net (Ronneberger et al., 2015). Its parameters are fully populated by the hypernetwork, offering greater flexibility, especially in estimating spatial derivatives (Bar-Sinai et al., 2019). Furthermore, the operator lives in a low-dimensional space (of dimension 384), far smaller than the size of a data, creating an information bottleneck (Tishby et al., 2000), that forces the model to focus on the most essential aspects of the dynamics, preventing overfitting to the initial conditions (level **(3)** variability above). DISCO is pretrained on two collections of PDE datasets. On PDEBench (Takamoto et al., 2022), it provides state-of-the-art performances on next frame prediction, with very few epochs. On The Well (Ohana et al., 2024), our model compares favorably to both standard transformer architectures, as well as meta-learning frameworks.

**Contributions.** Our key contributions are as follows: **(a)** To our knowledge, this is the first work to combine a transformer hypernetwork with a neural PDE solvers for spatiotemporal prediction. **(b)** We propose a model that tackles multi-physics-agnostic prediction with a tailored inductive bias for physical systems, providing improved interpretability by aligning more closely with classical numerical methods. **(c)** Our method achieves state-of-the-art performance on next-step prediction across multiple physical systems in the well-studied PDEBench dataset, requiring significantly fewer epochs than baseline models, and **(d)** achieves better numerical accuracy on multi-step rollouts compared to both standard transformer models and meta-learning frameworks. **(e)** Our model is the first to be scaled to such a diverse range of PDEs in 1D, 2D, 3D, with spatial resolutions up to $1024, 512 \times 512, 64 \times 64 \times 64$ respectively. In contrast, most models tackling multi-physics-agnostics prediction are trained on at most 2D data at $128 \times 128$ resolution. Upon publication, we will open-source our implementation.

## 2. Related works

**Neural solvers and operators.** While classical PDE solvers remain the state-of-the-art for achieving high precision, neural network-based surrogate solvers (Long et al.,

*Table 1.* The datasets from PDEBench (Takamoto et al., 2022) and The Well (Ohana et al., 2024) used in this paper.

| DATASET NAME | PHYSICAL DIMENSION | # OF FIELDS | RESOLUTION (TIME) | RESOLUTION (SPACE) | BOUNDARY CONDITIONS |
|---|---|---|---|---|---|
| BURGERS | 1D | 1 | 200 | 1024 | PERIODIC |
| SHALLOW WATER EQUATION | 2D | 1 | 100 | $128 \times 128$ | OPEN |
| DIFFUSION-REACTION | 2D | 2 | 100 | $128 \times 128$ | NEUMANN |
| INCOMP. NAVIER-STOKES (INS) | 2D | 3 | 1000 | $512 \times 512$ | DIRICHLET |
| COMP. NAVIER-STOKES (CNS) | 2D | 4 | 21 | $512 \times 512$ | PERIODIC |
| ACTIVE MATTER | 2D | 11 | 81 | $256 \times 256$ | PERIODIC |
| EULER MULTI-QUADRANTS | 2D | 5 | 100 | $512 \times 512$ | PERIODIC / OPEN |
| GRAY-SCOTT REACTION-DIFFUSION | 2D | 2 | 1001 | $128 \times 128$ | PERIODIC |
| RAYLEIGH-BÉNARD | 2D | 4 | 200 | $512 \times 128$ | PERIODIC $\times$ DIRICHLET |
| SHEAR FLOW | 2D | 4 | 200 | $256 \times 512$ | PERIODIC |
| TURBULENCE GRAVITY COOLING | 3D | 6 | 50 | $64 \times 64 \times 64$ | OPEN |
| MHD | 3D | 7 | 100 | $64 \times 64 \times 64$ | PERIODIC |
| RAYLEIGH-TAYLOR INSTABILITY | 3D | 4 | 120 | $64 \times 64 \times 64$ | PERIODIC $\times$ PERIODIC $\times$ SLIP |
| SUPERNOVA EXPLOSION | 3D | 6 | 59 | $64 \times 64 \times 64$ | OPEN |

2018; Hsieh et al., 2019; Li et al., 2020; Gelbrecht et al., 2021; Kovachki et al., 2023; Verma et al., 2024; Liu-Schiaffini et al., 2024) have opened up new possibilities for inferring approximate solutions quickly for certain PDEs. However, they require training on samples from the same PDE. Some variants, such as Karniadakis et al. (2021); Kochkov et al. (2021), incorporate corrective terms to approximate trajectory dynamics, but still lack adaptability to the context, which is a key feature of our method. Symbolic regression (Lemos et al., 2023) separates trajectory dynamics inference from integration but struggles with high-dimensional data and handle large search spaces. One could bypass estimating dynamics from context; for instance, Herde et al. (2024) train a large operator to predict the next frame from the previous one across various physical systems. However, this requires retraining at inference for each unseen PDE to allow the model to determine the dynamics.

**Meta-learning strategies for dynamical systems.** Rather than solving each PDE separately, meta-learning leverages shared weights across tasks, yet existing methods face two key limitations. First, they often rely on additional information, such as explicit equation forms (Linial et al., 2021; Sun et al., 2025) or weak supervision from PDE coefficients or domain-specific quantities (Wang et al., 2022; Alesiani et al., 2022), while others assume prior knowledge of PDE symmetries (Mialon et al., 2023). Second, their adaptation mechanisms are often restrictive, e.g., affine predictability (Blanke & Lelarge, 2023) or limiting adaptation to only the first layer of an operator (Zhang et al., 2024). In contrast, DISCO outputs an operator solely from prior frames, without assumptions on the PDE. Certain meta-learning strategies for PDEs (Kirchmeyer et al., 2022; Koupaï et al., 2024) circumvent these limitations but were trained only on variability levels **(2)** and **(3)**, not across different PDE

classes (level **(1)** in the introduction). They also require significant retraining at inference, making them impractical for large-scale data at inference—Koupaï et al. (2024) tested their approach on 2D data up to $256 \times 256$ resolution.

**Transformers and in-context models.** Without explicitly integrating an operator, transformer models trained to predict the next frame from a context of previous frames via attention have been applied to multi-physics-agnostic prediction (Yang et al., 2023; Liu et al., 2023; Yang & Osher, 2024; McCabe et al., 2024; Hao et al., 2024; Cao et al., 2024; Serrano et al., 2024). These architectures operate in a latent (i.e., token) space and, since they directly output the next frame, require learning a decoder. As a result, they must tackle not only the already challenging multi-physics-agnostic prediction task but also a reconstruction task, which is highly non-trivial for physics. In contrast, DISCO processes frames directly in the sample-space.

## 3. Methodology

### 3.1. Preliminaries

Let $(t, x) \in \mathbb{R} \times \mathbb{R}^n$ denote time and space. We consider PDEs defined in a spatial domain $\Omega \subset \mathbb{R}^n$ for $u : \mathbb{R} \times \Omega \to \mathbb{R}^m$ of the form

$$\partial_t u_t(x) = g\big(u_t(x),\, \nabla_x u_t(x),\, \nabla_x^2 u_t(x),\, \dots\big), \quad (1)$$
$$\mathcal{B}[u_t(x)] = 0, \quad x \in \partial\Omega, \quad (2)$$
$$u_0(x) = u_{\text{init}}(x), \quad x \in \Omega, \quad (3)$$

where different $g$ corresponds to different physics (different equations and/or parameters) governing the system, and the next two equations correspond to the boundary and initial conditions. Here, $n$ is the spatial dimension, typically

ranging from 1 to 3 in physical systems, and $m$ is the number of physical variables described by the PDE. For notational simplicity, we assume the PDE is time-homogeneous (i.e., $g$ does not depend on $t$) though our methods can be straightforwardly extended to the time-inhomogeneous case. For PDEs with higher-order time derivatives, specifically of order $r$, we can redefine the function of interest as $\bar{u} = (u, \partial_t u, \ldots, \partial_t^r u)$, which allows us to rewrite the PDE in the form of Eq. 1. Given a spatiotemporal trajectory with *unknown* $g$, our goal is to predict the state $u_{t+1}$ using the preceding $T$ successive states $u_{t-T+1}, \ldots, u_t$, where $T$ is referred to as the *context length*. In practice, the observed $u_t$ is discretized over grid points (assumed fixed and uniformly-distributed in this paper) rather than being a continuous spatial function, yet we still use continuous notation.

**Finite difference and neural PDEs.** Finite difference method is a classical approach for numerically solving an explicitly given PDE by discretization on a grid. This connection forms a basis for leveraging translation-equivariant networks to approximate PDE solutions. To illustrate this relationship, consider the standard 2D diffusion equation $\partial_t u_t(x) = \beta \Delta u_t(x)$, $x \in [0,1]^2$, subject to periodic boundary conditions, where $\beta > 0$ is the diffusion coefficient. Let the state $u_t$ be discretized over a uniform grid with spacing $\Delta x$, denoted by $u_{t,i,j}$ for $0 \leq i,j \leq N$. Using a standard centered finite difference scheme, the Laplacian $\Delta$ can be approximated as $u_t \star \theta$, where $\theta = \beta/(\Delta x)^2 [0,1,0;1,-4,1;0,1,0]$ is a small $3 \times 3$ convolution filter, and $\star$ denotes the convolution operation with periodic padding. Defining $\tilde{f}_\theta(u) := u \star \theta$, we have $u_{t+1} \approx u_t + \int_t^{t+1} \tilde{f}_\theta(u_s)\, ds$, which indicates that when the physics is explicitly known, we only need a single convolution layer with a few parameters to evolve the solution $u_t$ accurately.

Extending this concept to machine learning, Bar-Sinai et al. (2019) proposed replacing these fixed convolutional coefficients with learnable parameters, resulting in improved accuracy on certain grid sizes. For more complex PDEs involving nonlinear dynamics, additional layers with nonlinear activation functions are necessary to capture the underlying effects (Long et al., 2018; Kochkov et al., 2021). This connection between finite difference methods and CNNs highlights that relatively small CNN architectures, compared to transformer-based models, can effectively represent temporal PDEs while preserving spatial translation equivariance. This perspective has been explored in previous work (Ruthotto & Haber, 2020) and recently applied to design architectures for solving PDEs when the governing physics are known (Liu et al., 2024). This connection holds primarily for continuous-time or small discrete-time steps. For larger time steps $\Delta t$, the exact solution to the above heat equation still retains a convolutional structure, with a

significantly larger effective receptive field. Thus, using a CNN $\tilde{f}_\theta$ to directly approximate the solution in the form of $u_{t+\Delta t}(x) = \tilde{f}_\theta(u_t)$, as proposed in (Wang et al., 2022), would require a much larger network with more parameters, challenging to train.

### 3.2. Method: DISCovering an Operator from context

**Our framework.** We propose a model called DISCO that combines the best of both worlds from transformer models and classical numerical schemes (see Fig. 1): the parameters $\theta \in \mathbb{R}^{d_1}$ of a small network $f_\theta$—referred to as the *operator network*—are generated from an ad-hoc model $\psi_\alpha$, referred to as a *hypernetwork* (Ha et al., 2016), which uses the context to estimate $\theta$, leading to:

$$
\begin{cases}
\hat{u}_{t+1} = u_t + \displaystyle\int_t^{t+1} f_\theta(u_s)\, ds\,, \\
\theta = \psi_\alpha\big(u_{t-T+1}, \ldots, u_t\big)\,,
\end{cases}
\tag{4}
$$

where $\alpha \in \mathbb{R}^{d_2}$ are learnable parameters, while $\theta$ are parameters predicted by $\psi$, with $d_1$ and $d_2$ being the sizes of the operator network and hypernetwork, respectively. This formulation significantly structures the predictor, which now has a convolutional structure aligned with Eq. 1: the spatial derivatives are necessarily approximated using the convolutional kernels of $f_\theta$.

**Numerical integration.** The integral from $t$ to $t+1$ in Eq. 4 is discretized using a third order adaptive Runge-Kutta method (Bogacki & Shampine, 1996). Gradient backpropagation is performed using an adjoint sensitivity method which scales linearly in the number of integration steps and has low memory cost (Chen et al., 2018). The solver uses at maximum 32 integration steps, but can use less steps, thus automatically adapting to the effective "speed" of the current trajectory. An alternative would be to fix a budget of 32 frames and let the solver predict longer in the future if possible. It would provide a cheap way of training on long rollouts at training time, which we leave for future work.

**Operator network architecture $f_\theta$.** The operator network $f_\theta$ in DISCO has to be fast (because it will be integrated) and low-dimensional (to enforce a bottleneck). We used a U-Net with 4 downsampling blocks, a MLP in the deepest layers and 4 upsampling blocks with their respective skip connections. The input channel dimension is mapped to 8 before any downsampling. In the deepest layer, it is increased to $128 = 8 \cdot 2^4$, and goes through a MLP with hidden dimension of 256. We used GeLU activation (Hendrycks & Gimpel, 2016) and group normalization (Wu & He, 2018) with 4 groups. For datasets with non-periodic boundary conditions, we use reflection-padding for the spatial convolution and we add a mask representing the boundary as

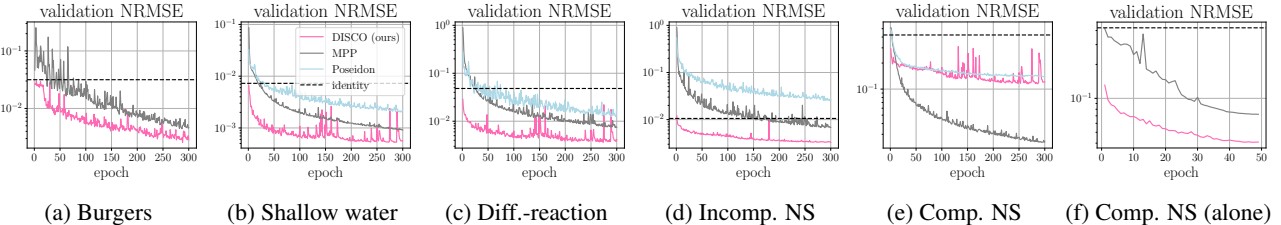

(a) Burgers    (b) Shallow water    (c) Diff.-reaction    (d) Incomp. NS    (e) Comp. NS    (f) Comp. NS (alone)

*Figure 2.* **Learning curves on PDEBench datasets.** DISCO achieves already decent predictions after one epoch, while an auto-encoder transformer architecture such as Poseidon or MPP requires many epochs before beating a simple identical prediction $\hat{u}_{t+1} = u_t$ (black dashed line). On shallow-water dataset for example, our model achieves $10^{-2}$ performance after one epoch, while Poseidon and MPP models require more than 150 epochs to reach same accuracy. This is an illustration of the better inductive bias implemented in our framework, which also avoids having to train a decoder.

additional channel as input to $f_\theta$. We end up with an operator network with roughly $d_1 \approx 200k$ parameters $\theta$ and have an intrinsic dimension of 384 as described below.

**Hypernetwork architecture $\psi$.** Multiple choices of hypernetwork could be used, but we decided to exploit transformers due to their favorable scaling properties (Lin et al., 2022). In fact, we emphasize that any type of meta learning approaches, like directly optimizing $\theta$ via gradient descent, could be considered yet it might not be computationally favorable. Our hypernetwork $\psi$ consists of three components: a CNN encoder that reduces spatial resolution, a processor composed of attention layers across time or space, and a final parameter generation block that outputs the parameters $\theta$ of the operator $f_\theta$. We refer the reader to App. C for details on the architecture. The output of the processor is averaged over both time and space, resulting in a single token of dimension 384, which is the intrinsic dimensionality of our operator network. Finally, we feed this token to a MLP with two hidden layers that progressively increases the channel dimension to recover the expected parameter shape of $\theta$. The output is then normalized so as to account for the different layers in the operator network $f_\theta$ and fed to a sigmoid to restrict the range of each parameter $\theta$, which is key in making the training stable. The resulting hypernetwork has roughly $d_2 \approx 120m$ parameters $\alpha$ when trained on PDEBench. We thus have $d_1 \ll d_2$, meaning that the size of the operator network $f_\theta$ is much smaller than the size of the hypernetwork $\psi$ (see Eq. 4). This is in order to limit overfitting and any memorization phenomenon by forcing the transformer hypernetwork to estimate the parameters of the trajectory rather than the next state.

**End-to-end training.** The two equations in Eq. 4 can be learned jointly in an end-to-end manner by solving the optimization problem

$$\min_\alpha \frac{1}{|\mathcal{D}|} \sum_{k \in \mathcal{D}} \text{Loss}(u_{t+1}^k, \hat{u}_{t+1}^k),$$

where each $k$ in the dataset $\mathcal{D}$ is a data context, that is a small trajectory, in the form of $(u_{t-T+1}^k, ..., u_t^k, u_{t+1}^k)$. In this paper, we choose the loss function as the normalized root mean square error (NRMSE); see App. D for the detailed definition. After the weights $\alpha$ are trained, predicting a specific dynamic from a given trajectory is a downstream task that can be tackled with a simple forward in our model, as no retraining is involved, unlike in standard meta-learning models such as (Koupaï et al., 2024).

## 4. Numerical experiments

**Multiple physics datasets.** In order to showcase the performances of DISCO on multi-physics-agnostic prediction, we consider two collections of datasets (see Tab. 1 and App A for details). The PDEBench dataset has been well studied in the literature with SOTA performances on multi-physics-agnostic prediction obtained by MPP (McCabe et al., 2024). The Well datasets (Ohana et al., 2024) are recently released datasets exhibiting many more examples of Physics evolution operators including 3D operators. Our study is one of the first to test a large model on this rich datasets. Both collections of datasets include simulations of PDEs with varying parameters and varying initial conditions. All contexts are kept at their original resolutions, except for the Rayleigh-Taylor instability dataset which originally figures $128 \times 128 \times 128$ and was downsampled to $64 \times 64 \times 64$ for memory purposes. Details on the equations, initial conditions, boundary conditions, and data generation can be found in App. A.

**Multiple physics training.** When trained on multiple physics, our model is made agnostic to the dataset name and only sees the spatial dimensionality of the data, 1D, 2D, 3D, the fields names, e.g., pressure, velocity and the type of periodic boundary conditions (see Tab. 1). Most of the weights in the hypernetwork $\psi$ are thus shared across different contexts, except for the first field embedding layer (McCabe et al., 2024), which contains field-specific weights, and the very last layer, which is learned per context dimension (1D, 2D, and 3D). The operator network $f_\theta$ hyperparameters, such as number of layers, number of downsamplings, hidden dimension, are unchanged across all the experiments

*Table 2.* **Comparison with SOTA model MPP on PDEBench datasets.** Next step prediction performance measured in NRMSE (lower is better). Our model trained for 300 epochs achieves SOTA performances set by the MPP model trained on 500 epochs on most datasets. Due to the smaller time resolution of the CNS dataset compared to the other PDEBench datasets (see Tab. 1), our model does not reach state-of-the-art performance. However, when trained on this dataset alone, DISCO does achieve better performance (see Fig. 2f for the training curves): 0.041 vs 0.072 after 50 epochs.

| Model | Epochs | Burgers | SWE | DiffRe2D | INS | CNS |
|---|---|---|---|---|---|---|
| MPP (retrained) | 300 | 0.0046 | 0.00091 | 0.0077 | 0.0068 | 0.037 |
| MPP (retrained) | 500 | 0.0029 | 0.00075 | 0.0058 | 0.0056 | 0.031 |
| MPP (McCabe et al., 2024) | 500 | - | 0.00240 | 0.0106 | - | **0.0227** |
| DISCO (ours) | 300 | **0.0027** | **0.00056** | **0.0047** | **0.0033** | 0.095 |

conducted in this paper. The operator network size will however vary in size with the context, e.g. if the context is 3D then the convolution layers are 3D layers and thus have more weights. Also, the first layer and last layer of the operator adapts to the fields present in the context. We refer the reader to App. C for further detail on the exact architectures and number of weights.

**Baselines.** In order to cover the different frameworks used in the literature to tackle multi-physics-agnostic prediction (see Sec. 2), we implemented the following three baselines

- the GEPS framework (Koupaï et al., 2024) is a meta-learning approach extending CODA (Kirchmeyer et al., 2022) which learns an operator $f_\theta$ on data from different contexts (called "environments" in their paper) by training decomposing $\theta$ into shared weights across the contexts and context-specific weights which need to be re-trained at inference. We use this framework on our U-Net operator $f_\theta$. Since we tackle multi-physics-agnostic prediction, the common weights are shared across all contexts, thus incorporating the three types of variability in our task.

- the Poseidon model (Herde et al., 2024) is based on a hierarchical multiscale vision transformer, trained on pairs of frames (i.e. contexts of a single frame) to predict a next frame in the future. This models also takes a continuous horizon variable $n$ to decide which future frame $u_{t+n}$ it predicts. It is thus an example of large operator network that is trained on multiple physics at the same time and must be finetuned, on unseen data, so as to estimate the new Physics.

- MPP (McCabe et al., 2024) is a transformer-based model which processes contexts of frames to directly predict the next frame. We did not implement (Serrano et al., 2024), which can be seen as an in-context enhanced model of MPP, since we did not focus on in-context capabilities in our paper.

On top of these models we also compared our results to 2 additional baselines (a large U-Net (Ronneberger et al.,

2015) and FNO (Li et al., 2020)) for a finetuning experiment presented later in the paper. See App. C.

**Next steps prediction performances.** Tab. 2 shows that our model beats state-of-the-art performances on next step prediction on PDEBench, reported in (McCabe et al., 2024), for most of the datasets. It is remarkable that only a few epochs are required to beat the performances on most of the datasets. As reported in Tab. 5, it only takes 35 epochs to beat the performances of a transformer model trained on 500 epochs. These results are confirmed by observing the validation loss along training (Fig. 2). As we can see, the Poseidon and MPP baselines, can take up 150 epochs to several epochs to beat a naive identity prediction (next step is predicted as the current step). This, we believe, is due to the fact that these baselines rely on an auto-encoding of the data, which is in itself a non-trivial task that requires long training. In contrast, DISCO does not employ a decoder, we learn to output the weights of an operator that acts in the data space directly. When trained jointly on the PDEBench datasets, we are do not compare favorably to MPP on the compressible Navier-Stokes dataset. This is due to the fact that this dataset has a much coarser time resolution (see Tab. 1) relatively to the other datasets. As a matter of fact, when trained on this dataset alone, DISCO outperforms a standard transformer, see Tab. 2 and Fig. 2f. In order for our model to better adapt to contexts with diverse "speeds" of evolution, one could imagine harmonizing the temporal resolution at the beginning of the training and then increasing it slowly for the contexts allowing it, which we leave for future work.

Tab. 6 assesses the performances of the models on predicting the next steps $t + n$ for $n \geq 1$. At training time, all the models were trained on predicting only one step in the future $t + 1$, except Poseidon which was trained on pairs $(t, t + n)$ as in (Herde et al., 2024). As expected this model performs decently on rollouts. As we can see, DISCO remains competitive on predicting future steps on PDE data. Note that improving the rollout performance of an autoregressive model is an active area of research (McCabe et al., 2023), and one could complement our model with techniques such as noise injection (Hao et al., 2024), among others.

*Table 3.* **Rollout performance of models trained on the Well datasets.** All models are trained for 50 epochs. We use the NRMSE (lower is better). DISCO, which relies on an operator and does not require learning a decoder, achieves state-of-the-art performances on a large-scale and diverse dataset as the Well (Ohana et al., 2024). See Fig. 5 for learning curves across training, and Figs. 10, 11, 12, 13 for rollout examples.

| MODEL | NO RETRAINING AT INFERENCE | DECODER FREE | TEST DATASET | NRMSE | | | |
|---|---|---|---|---|---|---|---|
| | | | | $t+1$ | $t+4$ | $t+8$ | $t+16$ |
| GEPS | ✗ | ✓ | ACTIVE MATTER | 0.436 | 0.852 | 0.878 | 0.894 |
| | | | EULER MULTI-QUADRANTS | 0.759 | 0.873 | 0.920 | 0.939 |
| | | | GRAY-SCOTT REACTION-DIFFUSION | 0.158 | 0.480 | 0.615 | 0.718 |
| | | | RAYLEIGH-BÉNARD | >1 | >1 | >1 | >1 |
| | | | SHEAR FLOW | >1 | >1 | >1 | >1 |
| | | | TURBULENCE GRAVITY COOLING | 0.509 | 0.961 | >1 | >1 |
| | | | MHD | 0.977 | 0.949 | 0.979 | 0.992 |
| | | | RAYLEIGH-TAYLOR INSTABILITY | >1 | >1 | >1 | >1 |
| | | | SUPERNOVA EXPLOSION | >1 | >1 | >1 | >1 |
| MPP | ✓ | ✗ | ACTIVE MATTER | 0.134 | 0.442 | 0.778 | >1 |
| | | | EULER MULTI-QUADRANTS | 0.070 | 0.166 | 0.285 | 0.446 |
| | | | GRAY-SCOTT REACTION-DIFFUSION | 0.0652 | 0.118 | 0.197 | 0.348 |
| | | | RAYLEIGH-BÉNARD | 0.115 | 0.243 | 0.396 | **0.636** |
| | | | SHEAR FLOW | 0.0522 | 0.138 | 0.261 | 0.501 |
| | | | TURBULENCE GRAVITY COOLING | 0.254 | 0.460 | 0.670 | 0.938 |
| | | | MHD | 0.435 | **0.655** | **0.876** | >1 |
| | | | RAYLEIGH-TAYLOR INSTABILITY | 0.366 | 0.506 | 0.720 | **0.995** |
| | | | SUPERNOVA EXPLOSION | 0.673 | 0.803 | 0.913 | >1 |
| DISCO (OURS) | ✓ | ✓ | ACTIVE MATTER | **0.114** | **0.367** | **0.664** | **0.995** |
| | | | EULER MULTI-QUADRANTS | **0.0637** | **0.157** | **0.248** | **0.401** |
| | | | GRAY-SCOTT REACTION-DIFFUSION | **0.00990** | **0.0373** | **0.0789** | **0.215** |
| | | | RAYLEIGH-BÉNARD | **0.0741** | **0.208** | **0.369** | 0.758 |
| | | | SHEAR FLOW | **0.0249** | **0.0846** | **0.157** | **0.307** |
| | | | TURBULENCE GRAVITY COOLING | **0.165** | **0.418** | **0.605** | **0.822** |
| | | | MHD | **0.298** | 0.684 | 0.905 | >1 |
| | | | RAYLEIGH-TAYLOR INSTABILITY | **0.0789** | **0.259** | **0.458** | >1 |
| | | | SUPERNOVA EXPLOSION | **0.387** | **0.656** | **0.778** | **0.982** |

While PDEBench (Takamoto et al., 2022) is an early dataset designed for evaluating prediction models in Physics, the Well dataset (Ohana et al., 2024) is a unified collection of more diverse scenarios covering biological systems, fluid dynamics, including astrophysics simulations. The challenge lies in the diverse operators in the data, making it unclear whether a model can be trained on such varied physics. Tab. 3 reports the results of DISCO on the Well, as well as two of our baselines (Poseidon was not implemented for 3D data, nor was it for non-square 2D data) for model trained on 50 epochs. As we can see, we compare favorably on all the datasets we trained on, especially on rollouts: incorporating the right inductive biases with an operator and avoiding to train an auto-encoder enable to beat standard spatiotemporal prediction frameworks for PDEs. To date, we are one of the first paper training jointly on such diverse dataset of evolution operators in Physics.

Let us note that the GEPS adaptation framework does not manage to obtain competitive performances, at least when

applied on our operator network. This framework learns shared parameters across the various context during training and then adapts these parameters through gradient descent at inference time. While this framework seems competitive on contexts originating from the same class of PDE (Koupaï et al., 2024), our hypothesis is that on a multi-physics-agnostic prediction on large-scale data it is too difficult in practice to learn shared parameters of a relatively "small" operator $f_\theta$ on different classes of PDEs. Thus, what we show in this paper is that it is possible to learn shared parameters, but as the ones of the hypernetwork, not as parameters of the operator itself.

**Information bottleneck and operator space.** DISCO employs a low-dimensional operator $f_\theta$ (see Eq. 4) with an intrinsic dimension of 384, very much smaller than the typical input sizes. We study the impact of this information bottleneck. When trained on multiple datasets, the parameters $\theta$ returned by our hypernetwork depend on both the initial conditions and the PDE governing the given context.

| TRUTH | U-Net | FNO | GEPS | Poseidon | MPP | DISCO (ours) |

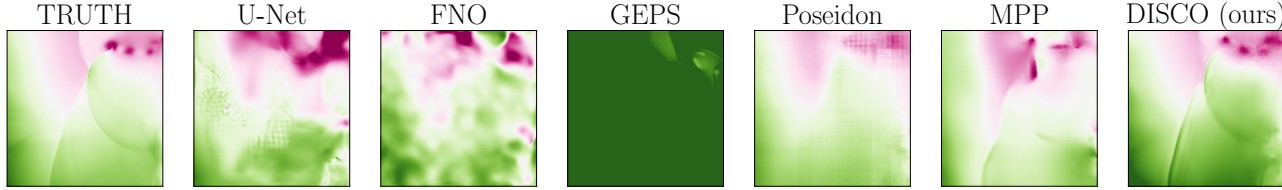

*Figure 3.* Step $t + 16$ predictions for different models on the Euler dataset with fixed PDE coefficients. U-net and FNO are trained from scratch while GEPS, Poseidon, MPP and DISCO are finetuned from pretrained models on PDEBench (see also Tab. 4).

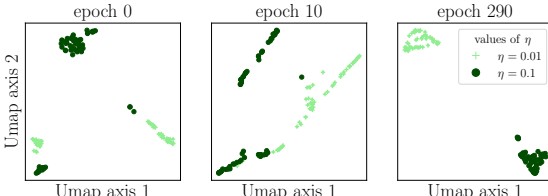

*Figure 4.* Visualization of the parameter space using UMAP on compressible Navier-Stokes data. Each point represents a set of parameters $\theta$ predicted by the hypernetwork $\psi$ for a given context. The hypernetwork tends to predict similar $\theta$ values when the context is derived from PDEs with the same parameters $\eta$, but different initial conditions. This demonstrates the generalization capability of our approach to varying initial conditions.

We show that our framework is capable of reducing the variability introduced by the initial condition, allowing it to focus primarily on the PDE dynamics. For that, we study the parameter space $\theta$ for certain 2D contexts, through dimensionality reduction. The size of $\theta$ varies slightly from a context to the other. This is because the first and last layer of $f_\theta$, which is a U-Net, adapt to the number of channels in the data. The remaining parameters have the same size 157 544 across all 2D contexts, and we perform dimensionality reduction via Umap (McInnes et al., 2018) on them, for 128 contexts from compressible Navier-Stokes dataset, with two different shear viscosity $\eta = 0.01$ and $\eta = 0.1$ (see App. A). Fig. 4 visualizes the parameter space across different stages of the training, showing that our model progressively identifies two distinct clusters, corresponding to the two physical parameter values. Thus, the hypernetwork clusters together contexts originating from the same PDE coefficient, despite different initial conditions, which is the key for generalizing to initial conditions and reducing unnecessary variability of level **(3)** (see Sec. 1).

**Fine-tuning on unseen physics.** We further assess the generalization properties of DISCO to an unseen PDE with unseen initial conditions. Specifically, we fine-tune a GEPS model, the Poseidon model and the MPP model, both initially trained on PDEBench, on the Euler dataset with a single set of parameters so that this unseen Physics exhibits only one operator in the data. The dataset is governed by compressible Navier-Stokes equations and contains initial

*Table 4.* **Finetuning on unseen Physics.** Performances in NRMSE of pretrained models, GEPS, Poseidon, MPP, when fine-tuned on the unseen Euler equations dataset with a single, fixed, set of coefficients ($\gamma = 1.4$) after 20 epochs of finetuning. For comparison, we show the performances of two other models: U-Net, FNO, trained from scratch.

| UNET | FNO | GEPS | POSEIDON | MPP | DISCO |
|---|---|---|---|---|---|
| 0.050 | 0.067 | 0.36 | 0.052 | 0.032 | **0.029** |

conditions not encountered during pretraining. Since there is only one fixed PDE in the data, we also compare against networks such as U-Net (Ronneberger et al., 2015) (not to be confused with the smaller U-Net $f_\theta$ we used), Fourier Neural Operators (FNO) (Li et al., 2020), and autoregressive-diffusion models(Kohl et al., 2024), which are designed to learn a fixed operator from data. Tab. 4 and Fig. 3 show our model achieves the best performance when fine-tuned on this new dataset, demonstrating its ability to generalize to a new PDE and novel initial conditions better than a standard transformer (due to better inductive biases) and other neural operator methods (thanks to pretraining).

**Ablation on model size.** Our model size depends both on the size of the operator network $f_\theta$ which is integrated to predict future frames, and the size of the hypernetwork $\psi$ which predict weights $\theta$ from the context. Tab. 7 shows variations in the performances with changes on both networks. The model is trained only on the diffusion-reaction dataset from PDEBench.

## 5. Conclusion

We introduce DISCO, a general and efficient framework for multi-physics-agnostic prediction. It uses a transformer-based hypernetwork that adapt to varying contexts to generate the parameters of a neural PDE solver, which leverages continuous-time dynamics and spatial translation equivariance. Compared to pure transformers, DISCO achieves superior generalization and fine-tuning performance. The operator network in our framework, implemented using convolutions, is primarily inspired by finite difference schemes on a uniform mesh. However, many challenging problems

in physics involve non-uniform meshes or arbitrary geometries. In such cases, finite volume or finite element schemes can be implemented using graph neural networks instead (Pfaff et al., 2020; Zhou et al., 2022; Brandstetter et al., 2022; Zhou et al., 2025). In particular, several papers, such as (Esmaeilzadeh et al., 2020; Lino et al., 2022; Cao et al., 2023), propose U-Net-like graph neural network architectures, which are natural candidates for our operator class.

Promising directions for expanding the capabilities of DISCO include multi-step rollout during training, that is, predicting future steps (e.g., $t + 1, t + 2, t + 3$) with the same neural solver and backpropagating from a loss averaged on all of them. This is made particularly advantageous in DISCO as applying the small neural solver requires little memory. While our method is validated on time-independent dynamics, extending it to time-dependent systems requires integrating temporal inputs into our hypernetwork and is left for future work. Additionally, our convolutional operator network is inspired by finite difference schemes; exploring spectral methods could lead to architectures similar to FNOs (Li et al., 2020), leveraging new inductive biases.

## Acknowledgement

We thank our close colleagues from Polymathic AI for their valuable feedback on our paper, including Michael McCabe, Ruben Ohana, Lucas Meyer, Miles Cranmer, François Lanusse, Bruno Régaldo-Saint Blancard, Alberto Bietti, and Shirley Ho. We also thank Patrick Gallinari and Lawrence Saul for insightful discussions, as well as Louis Serrano and Armand Kassaï Koupaï. We also thank Géraud Krawezik and the Scientific Computing Core at the Flatiron Institute, a division of the Simons Foundation, for providing computational resources and support. Part of this work was supported by PEPR IA on grant SHARP ANR-23-PEIA-0008.

## Impact Statement

This paper presents work whose goal is to advance the field of Machine Learning. There are many potential societal consequences of our work, none of which we feel must be specifically highlighted here.

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

# A. Dataset description

For sake of self-consistency, we provide the description of the datasets considered in this paper (see Tab. 1) and which originate from two benchmark papers (Takamoto et al., 2022; Ohana et al., 2024), along with their underlying PDEs. The corresponding data is publicly available.

We used 5 datasets from PDEBench, comprising the 2D shallow-water equations, 2D diffusion-reaction, 2D incompressible and compressible Navier-Stokes datasets. These are the datasets used to train MPP (McCabe et al., 2024). In addition, we added the 1D Burgers equations dataset in order to showcase our model can be trained on 1D and 2D data jointly.

We used 9 datasets from the Well, comprising the 2D active matter, 2D Euler multi-quadrants, 2D Gray-Scott reaction-diffusion, 2D Rayleigh-Bénard, 2D shear flow, 3D turbulence gravity cooling, 3D MHD, 3D Rayleigh-Taylor instability and 3D supernova explosion datasets. We initially excluded some of the datasets present in the Well, such as the convective envelope red supergiant dataset, or the post neutron star merger dataset, which are not stored using cartesian coordinate systems. We provide below the description of the shear flow and Euler multi-quadrants datasets which were used for specific additional experiments in this paper and refer the reader to (Ohana et al., 2024) for the description of the other datasets.

Examples of trajectories are provided in the rollout Fig. 7, 8, 9 for PDEBench and Fig. 10, 11, 12, 13 for The Well.

## A.1. Burgers equations

The Burgers equations model the evolution of a 1D viscous fluid. They combine a nonlinear advection term with a linear diffusion term

$$\partial_t u_t + u_t \partial_x u_t = \frac{\nu}{\pi} \partial^2_{xx} u_t$$

where $\nu$ is the diffusion coefficient.

The boundary conditions are set to periodic. The dataset is part of PDEBench (Takamoto et al., 2022) and was generated using a temporally and spatially 2nd-order upwind difference scheme for the advection term, and a central difference scheme for the diffusion term. We refer to Fig. 7 for examples of trajectories.

## A.2. Shallow water equations

Shallow-water equations, derived from Navier-Stokes equations present a suitable framework for modeling free-surface flow problems. They take the form of a system of hyperbolic PDEs

$$\partial_t h + \partial_x h + \partial_y hv = 0\,,$$

$$\partial_t hu + \partial_x(u^2 h + \frac{1}{2}g_r h^2) + \partial_y uvh = -g_r h \partial_x b\,,$$

$$\partial_t hv + \partial_y(v^2 h + \frac{1}{2}g_r h^2) + \partial_x uvh = -g_r h \partial_y b\,,$$

where $h$ is the water depth, $u, v$ are the velocities in horizontal and vertical direction, $b$ is the bathymetry field, and $g_r$ describes the gravitational acceleration.

The initial state is a circular bump in the center of the domain. The dataset is part of PDEBench (Takamoto et al., 2022) and was generated using a finite volume solver (Ketcheson et al., 2012). We refer to Fig. 7 for an example trajectory.

## A.3. Diffusion reaction equations

A 2D diffusion-reaction equation models how a substance spreads and reacts over time, capturing the combined effects of diffusion and chemical or biological reactions in two dimensions

$$\partial_t u = D_u \partial^2_{xx} u + D_u \partial_{yy} u + R_u\,,$$

$$\partial_t v = D_v \partial^2_{xx} v + D_v \partial_{yy} u + R_v\,,$$

where $D_u, D_v$ are the diffusion coefficients for the activator $u$ and inhibitor $v$ and $R_u, R_v$ are the respective reaction functions, which takes the form $R_u(u,v) = u - u^3 - k - v$ and $R_v(u,v) = u - v$ where $k$ is a constant. The

initial states are random Gaussian white noises. The problem involves no-flow Neumann boundary condition, that is $D_u\partial_x u = 0, D_v\partial_x v = 0, D_u\partial_y u = 0, D_v\partial_y v = 0$ on the edges of the square domain. The dataset is part of PDEBench (Takamoto et al., 2022) and was generated using a finite volume method as spatial discretization and fourth-order Runge-Kutta method as time discretization. We refer to Fig. 8 for examples of trajectories.

### A.4. Incompressible Navier-Stokes

This dataset considers a simplification of Navier-Stokes equation that writes

$$\nabla \cdot \mathbf{v} = 0 \ , \ \ \rho(\partial_t\mathbf{v} + \mathbf{v} \cdot \nabla\mathbf{v}) = -\nabla p + \eta\Delta\mathbf{v} + \mathbf{u}$$

where $\mathbf{v}$ is the velocity vector field, $\rho$ is the density and $\mathbf{u}$ is a forcing term and $\nu$ is a constant viscosity.

Initial states and forcing term $\mathbf{u}$ are each drawn from isotropic random fields with a certain power-law power-spectrum. The boundary conditions are Dirichlet, imposing the velocity field to be zero at the edges of the square domain. The dataset is part of PDEBench (Takamoto et al., 2022) and was generated using a differentiable PDE solver (Holl et al., 2020). We refer to Fig. 8 for examples of trajectories.

### A.5. Compressible Navier-Stokes

The compressible Navier-Stokes describe the motion of a fluid flow

$$\partial_t\rho + \nabla \cdot (\rho\mathbf{v}) = 0$$
$$\rho(\partial_t\mathbf{v} + \mathbf{v} \cdot \nabla\mathbf{v}) = -\nabla p + \eta\Delta\mathbf{v} + (\zeta + \eta/3)\nabla(\nabla \cdot \mathbf{v}),$$
$$\partial_t(\epsilon + \frac{1}{2}\rho v^2) + \nabla \cdot ((\epsilon + p + \frac{1}{2}\rho v^2)\mathbf{v} - \mathbf{v} \cdot \sigma') = 0$$

where $\rho$ is the density, $\mathbf{v}$ is the velocity vector field, $p$ the pressure, $\epsilon = p/(\Gamma - 1)$ is the internal energy, $\Gamma = 5/3$, $\sigma'$ is the viscous stress tensor, and $\eta, \zeta$ are the shear and bulk viscosity. The boundary conditions are periodic.

The dataset is part of PDEBench (Takamoto et al., 2022) and was generated using 2nd-order HLLC scheme (Toro et al., 1994) with the MUSCL method (Van Leer, 1979) for the inviscid part, and the central difference scheme for the viscous part. This dataset contains trajectories with only 5 steps into the future and was used solely for training.

We refer to Fig. 9 for an example of trajectory.

### A.6. Shear flow

This phenomenon concerns two layers of fluid moving in parallel to each other in opposite directions, which leads to various instabilities and turbulence. It is governed by the following incompressible Navier-Stokes equation

$$\frac{\partial u}{\partial t} - \nu \Delta u + \nabla p = -u \cdot \nabla u.$$

where $\Delta = \nabla \cdot \nabla$ is the spatial Laplacian, with the additional constraints $\int p = 0$ (pressure gauge). In order to better visualize the shear, we consider a passive tracer field $s$ governed by the advection-diffusion equation

$$\frac{\partial s}{\partial t} - D \Delta s = -u \cdot \nabla s.$$

We also track the vorticity $\omega = \nabla \times u = \frac{\partial u_z}{\partial x} - \frac{\partial u_x}{\partial z}$ which measures the local spinning motion of the fluid. The shear is created by initializing the velocity $u$ at different layers of fluid moving in opposite horizontal directions. The fluid equations are parameterized by different viscosity $\nu$ and tracer diffusivity $D$.

The data was generated using an open-source spectral solver (Burns et al., 2020) with a script that is publicly available. Refer to Fig. 12 for an example trajectory.

### A.7. Euler multi-quadrants (compressible)

Euler equations are a simplification of Navier-Stokes in the absence of viscosity

$$\partial_t U + \partial_x F(U) + \partial_y G(U) = 0,$$

with

$$U = \begin{pmatrix} \rho \\ \rho u \\ \rho v \\ e \end{pmatrix} \ , \ F = \begin{pmatrix} \rho u \\ \rho u^2 + p \\ \rho uv \\ u(e + p) \end{pmatrix} \ , \ G = \begin{pmatrix} \rho v \\ \rho uv \\ \rho v^2 + p \\ v(e + p) \end{pmatrix}$$

where $\rho$ is the density, $u, v$ are the horizontal and vertical velocities, $p$ is the pressure and $e$ is the energy defined by

$$e = \frac{p}{\gamma - 1} + \frac{1}{2}\rho(u^2 + v^2) \,.$$

The initial state is a piecewise constant signal composed of quadrants, which then evolves in multi-scale shocks. We refer to Fig. 10 for an example trajectory.

## B. Benchmark models hyperparameters

In this work, we compared DISCO with 5 baselines, a U-net (Ronneberger et al., 2015), a Fourier neural operator (Li et al., 2020) implemented using `neuralop` (Kovachki et al., 2023), GEPS (Koupaï et al., 2024) which is a meta-learning framework for operator learning, Poseidon (Herde et al., 2024), which is a transformer operator and MPP, which is a transformer based on axial attention (McCabe et al., 2024). We expose here the hyperparameters used in the paper for these models. The number of parameters provided for each model are the ones for the pretraining on PDEBench (see Tab. 2) or, if not applicable (U-Net, FNO), are the ones of the models trained on the Euler multi-quadrants dataset (see Tab. 4).

**U-Net.** We considered a standard U-Net (Ronneberger et al., 2015) with 4 down-(and up) sampling blocks, spatial filters of size 3 and initial dimension of 48. The resulting model has 17M learnable parameters. Note that this U-Net model is not to be confused with the operator network $f_\theta$ we used in DISCO which is much smaller (around 200k parameters).

**FNO.** We considered a standard Fourier neural operator (Li et al., 2020) with 4 Fourier blocks, spectral filters of size 16 (number of Fourier modes), and 128 hidden dimensions. The resulting model has 19M learnable parameters.

**GEPS.** This meta-learning framework (Koupaï et al., 2024) proposes to learn an operator $f_\theta$ by decomposing $\theta$ into shared parameters across different contexts and context-specific ("environment-specific" in their paper). The learnable parameters are the shared parameters, the parameters of the linear hypernetwork which process a "code" $c$ for each context, and the codes themselves. A code size is chosen in (Koupaï et al., 2024) to be at least the number of parameters in the underlying PDE. We chose the codes to be of dimension 12 in our case. This meta-learning strategy is performed on our class of U-Net operator $f_\theta$ which is of size roughly $200k$. The total number of trainable parameters in the shared weights and the hypernetwork is around 400k. Given our multi-physics-agnostic task (see Sec. 1) we cannot assume we know the specific dataset a context is originated from. So that, when trained on a collection of datasets, instead of sharing weights per dataset, which would lead to one vector of shared parameters for compressible Navier-Stokes, one vector for Burgers etc ... to comply with the task tackled in this paper and to yield a fair comparison, we chose to share one vector of parameters across all the contexts, thus facing the three levels of variability **(1)**, **(2)**, **(3)** (see Sec. 1). At inference, all the parameters are frozen, and only the codes are optimized. This optimization is done with a learning rate that is tuned on the validation set. We found that the optimal learning rate for optimizing a code on a single context was $10^{-2}$ for PDEBench and $10^{-1}$ for The Well (we tested three values $10^{-3}, 10^{-2}, 10^{-1}$).

**MPP.** We considered MPP, which is an axial vision transformer (McCabe et al., 2024) with patch size of 16, 12 attention layers, 12 attention heads per layer, 768 hidden dimensions. The resulting model has 158M learnable parameters. The training

## C. Additional details on DISCO architecture

### C.0.1. OPERATOR NETWORK

The operator network $f_\theta$ in DISCO is a Unet with 4 subsampling layers, starting channel dimension of 8. It is first composed of two convolutions with kernel 3 operating on 8 channels. Then followed by the downsampling blocks. At the lowest

resolution (the bottleneck), we apply a MLP with two layers, going from 128 to 256 channels and going back to 128 channels. A downsampling block is composed of a first convolution with kernel size 2 and stride 2 followed by a GeLU activation and a convolution of kernel size 3. An upsampling block is first composed of a transpose convolution with kernel of size 2 followed by a GeLU activation and a convolution of kernel size 3. The skip connections between layers of comparable resolution are performed by concatenation. The resulting architecture has roughly 200k parameters.

While most recent U-Net architectures utilize double convolutions as well as ConvNeXt blocks, we could not allow to use them in order to reduce the number of coefficients in the architecture to the minimum so as to enforce a bottleneck that is justified in Sec. 3. We also tried adding attention layers in the bottleneck of our U-Net, however, the gain in performance was not worth the additional computational cost.

C.0.2. HYPERNETWORK

The hypernetwork is an axial vision transformer closely following the design of MPP (McCabe et al., 2024) (small version), except that it replaces the decoder by a MLP to generate the parameters of the operator network. It consists of three parts, a CNN encoder with a first field embedding layer, which is a linear transform that adapts to varying number of fields in the input (McCabe et al., 2024). Then, we apply three convolution layers of kernel sizes 4, 2, 2 respectively, with GeLU activation, resulting in an effective patch size of 16. The hidden dimension (token space) is 384. After the encoder, the processor cascades 12 time-space attention blocks, each containing a time attention, and axial attentions along each space dimension (McCabe et al., 2024). Each attention block contains 6 heads and uses relative positional encodings. After the attention blocks, we average over time and space to yield a unique token of dimension 384, This token goes through a MLP with 3 linear layers with intermediate dimensions of 384 and output dimension the dimension of the operator network: roughly 200k. The output, $\theta$ in Eq. 4, is going to populate the different layers of the operator $f_\theta$. These layers being of different size, their respective weights $(\theta_1, \theta_2 \ldots$ on Fig. 1) are expected to be of different norm. To enforce this, we apply a signed sigmoid $\sigma_{\text{signed}}(x) = 2\sigma(x) - 1$ with the following normalization:

$$\theta_i = N_i \lambda \sigma_{\text{signed}}\big(\widetilde{\theta}_i/(N_i\lambda)\big) \tag{5}$$

where $\widetilde{\theta}$ is the un-normalized output of the MLP, $i$ is the layer index in the operator network, $N_i$ is the norm that the weights should have at initialization (set by PyTorch) for this layer, and $\lambda$ is a factor that was arbitrarily set to 2 for allowing the training to gradually increase the norm of $\theta$ while constraining it to avoid instabilities.

# D. Training details

**Hyperparameters.** For joint training on multiple datasets, we considered the same hyperparameters than McCabe et al. (2024), with batch size of 8, gradient accumulation every 5 batches, epoch size of 2000 batches ("fake epochs"). For single model trainings, we considered batch size of 32 with no gradient accumulation. In certain cases where the optimization was unstable, in particular, when we tried using only 2 intermediate number of steps (see Tab. 7), we used gradient clipping, clipping the total norm of the gradients to a default norm of 1.0. The data is split into train, validation, and test sets with an 80%, 10%, and 10% division, respectively.

**Optimization.** All trainings were performed using the adaptive Nesterov optimizer (Xie et al., 2024) and a cosine schedule for the learning rate. Using AdamW optimizer with varying learning rate did not improve overall performance in the cases we tested. The large-scale experiments are run for a fixed number of 300 epochs. We used a weight decay of 0.001 and drop path of 0.1.

**Loss.** A normalized root mean square error (NRMSE) is used for both monitoring the training of the model and assessing the performances in this paper. For two tensors $u$ (target) and $\hat{u}_{t+1}$ (prediction) with $C$ channels

$$\text{Loss}(u, \hat{u}) = \frac{1}{C} \sum_{c=1}^{C} \frac{\|u^c - \hat{u}^c\|_2}{\|u^c\|_2 + \epsilon} \tag{6}$$

where the $\ell^2$ norm $\|\cdot\|_2$ is averaged along space and $\epsilon = 10^{-7}$ is a small number added to prevent numerical instabilities. For a batch of data, this loss is simply averaged.

**Software.** The model trainings were conducted using python v3.11.7 and the PyTorch library v2.4.1 (Paszke et al., 2019).

*Table 5.* **Number of epochs to reach SOTA performance on PDEBench datasets.**

| Model | # parameters | Burgers | SWE | DiffRe2D | INS | CNS | CNS (alone) |
|---|---|---|---|---|---|---|---|
| MPP (retrained) | 160m | 500 | 500 | 500 | 500 | **500** | 50 |
| DISCO (ours) | 119m | **277** | **70** | **55** | **35** | - | **7** |

*Table 6.* **Rollout performance of models trained on PDEBench datasets.** All models are trained for 300 epochs. We use the NRMSE (lower is better). DISCO which relies on learning an operator explicitly, without having to learn a decoder, achieves better performance on next step prediction ($t+1$). Compared to Poseidon (Herde et al., 2024), which is trained on multiple future step prediction, our model, trained only on the next step ($t+1$) remains competitive. Best performance on each dataset is in bold, second best is underscored.

| MODEL | NO RETRAINING AT INFERENCE | DECODER FREE | TEST DATASET | NRMSE | | | | |
|---|---|---|---|---|---|---|---|---|
| | | | | $t+1$ | $t+4$ | $t+8$ | $t+16$ | $t+32$ |
| GEPS | ✗ | ✓ | BURGERS | 0.508 | 0.777 | 0.660 | 0.629 | 0.567 |
| | | | SWE | 0.0720 | 0.139 | 0.238 | 0.525 | 0.860 |
| | | | DIFFRE2D | 0.927 | >1 | >1 | >1 | >1 |
| | | | INS | 0.794 | 0.947 | 0.986 | 0.987 | 0.986 |
| | | | CNS | 0.604 | 0.784 | - | - | - |
| POSEIDON | ✗ | ✗ | BURGERS | - | - | - | - | - |
| | | | SWE | 0.00207 | **0.00207** | **0.00237** | **0.00331** | 0.0380 |
| | | | DIFFRE2D | 0.0141 | 0.0154 | **0.0227** | **0.0379** | **0.0821** |
| | | | INS | 0.0250 | 0.0318 | 0.0479 | 0.0869 | 0.211 |
| | | | CNS | 0.135 | 0.219 | - | - | - |
| MPP | ✓ | ✗ | BURGERS | 0.00556 | 0.0242 | 0.0526 | 0.107 | 0.191 |
| | | | SWE | 0.00145 | 0.00330 | 0.00592 | 0.0112 | 0.0327 |
| | | | DIFFRE2D | 0.0109 | 0.0283 | 0.0535 | 0.109 | 0.238 |
| | | | INS | 0.00936 | 0.0274 | 0.0557 | 0.132 | 0.379 |
| | | | CNS | **0.0349** | **0.0748** | - | - | - |
| DISCO (OURS) | ✓ | ✓ | BURGERS | **0.00347** | **0.0130** | **0.0298** | **0.0686** | **0.163** |
| | | | SWE | **0.000648** | 0.00226 | 0.00384 | 0.00554 | **0.00951** |
| | | | DIFFRE2D | **0.00448** | **0.0142** | 0.0291 | 0.0501 | 0.0832 |
| | | | INS | **0.00347** | **0.0139** | **0.0277** | **0.0556** | **0.113** |
| | | | CNS | 0.107 | 0.208 | - | - | - |

**Hardware.** All model trainings were conducted using Distributed Data Parallel across 4 or 8 Nvidia H100-80Gb GPUs.

# E. Additional experiments

This section contains additional experiments on the models used in this paper.

**Translation equivariance in standard transformers.** As a concrete example of the issues encountered by a transformer-based architecture, which does not naturally encode translation equivariance, let us consider shifting all the states in a context, as well as the following state, by a vector $v$ of increasing norm $\|v\|$. Fig. 6 shows that the transformer architecture quickly struggles to predict the next state, although it is competitive when no translation is applied, $\|v\| = 0$.

**Additional rollout examples.** Figs. 7, 8, 9, show additional rollout trajectories for the PDEBench datasets. Figs. 10, 11, 12, 13 are showing rollout trajectories for The Well datasets.

*Table 7.* Influence of the model size on the accuracy, on the diffusion-reaction dataset from PDEBench. Each table corresponds respectively to variations over $\alpha$ and $\theta$. Gray rows indicate the default values chosen in the paper.

| hyperparameters | | | num. weights $\alpha$ | NRMSE |
|---|---|---|---|---|
| num. layers | num. heads | dim | | |
| 4 | 3 | 192 | 65m | 0.00211 |
| 12 | 6 | 384 | 101m | 0.00133 |
| 12 | 12 | 768 | 280m | 0.00203 |

| hyperparameter $c_{start}$ | num. parameters $\theta$ | NRMSE |
|---|---|---|
| 4 | 41k | 0.00330 |
| 8 | 158k | 0.00133 |
| 12 | 353k | 0.000971 |

(a) Active matter

(b) Euler

(c) Gray-Scott

(d) Rayleigh-Bénard

(e) Shear flow

(f) Turbulence gravity cooling

(g) MHD

(h) Rayleigh-Taylor instability

(i) Supernova explosion

*Figure 5.* **Learning curves on The Well datasets.** DISCO achieves better performance on next step prediction compared to a MPP model retrained from scratch.

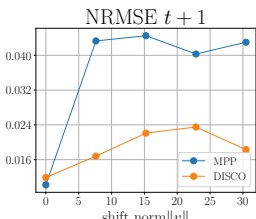

*Figure 6.* Performance on the shearflow dataset when context and target are shifted by $v \in \mathbb{R}^2$. While the transformer performs well at $v = 0$, it declines more than DISCO under shifts due to the lack of an inductive bias for translation equivariance.

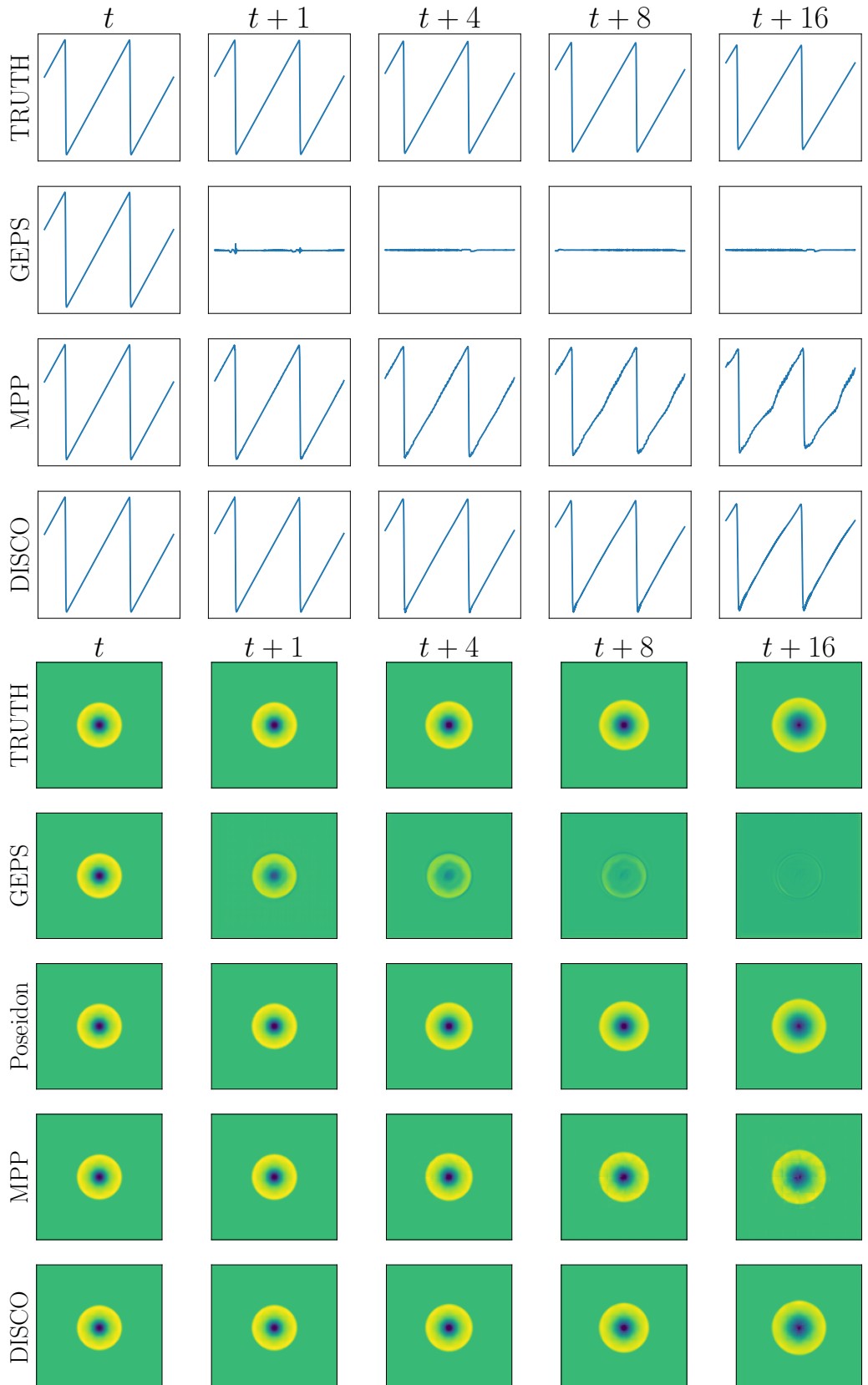

*Figure 7.* Examples of rollout trajectory for the Burgers (top) and shallow-water (bottom) datasets from PDEBench.

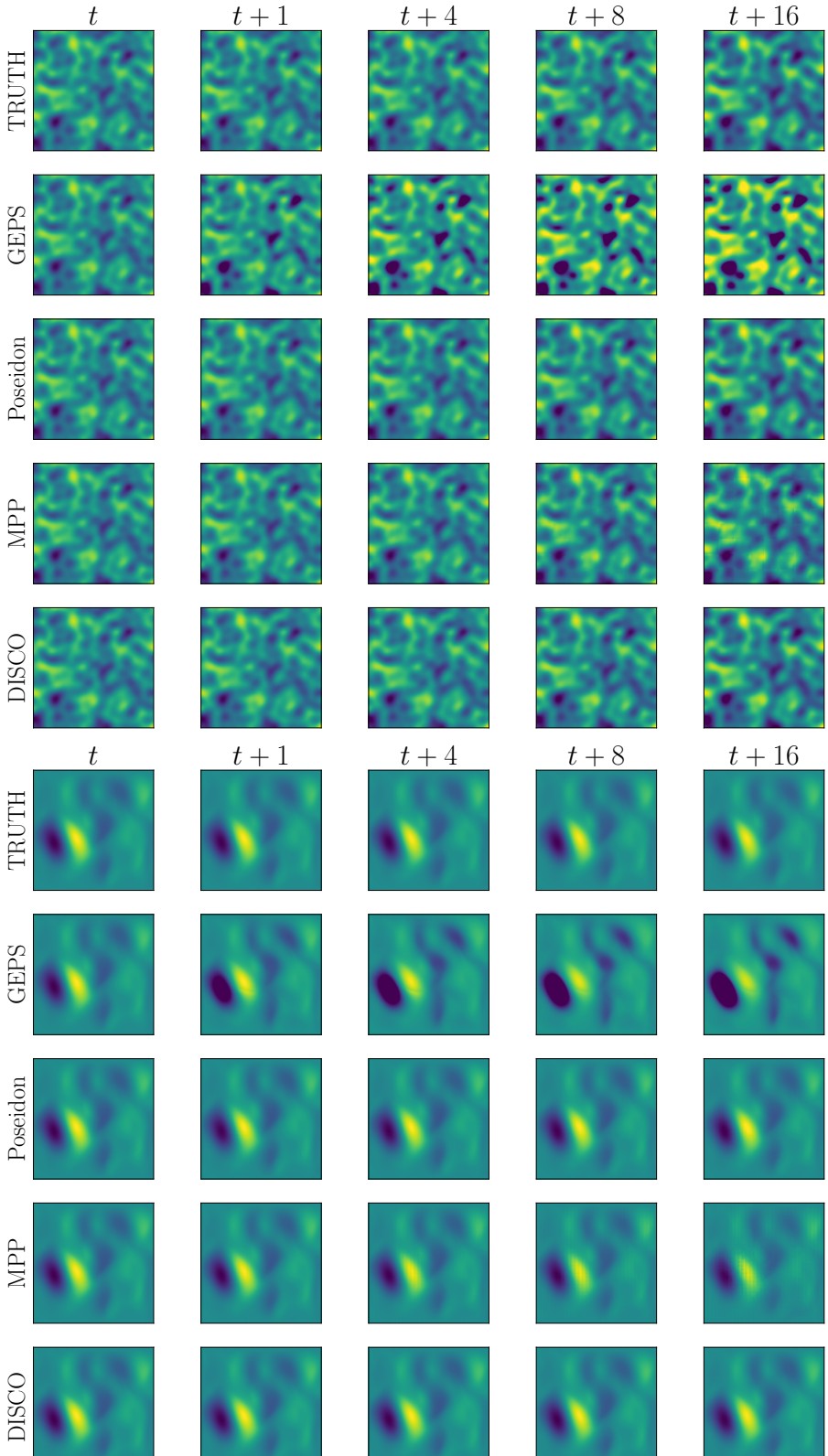

*Figure 8.* Examples of rollout trajectory for the diffusion-reaction (top) and incompressible Navier-Stokes datasets from PDEBench.

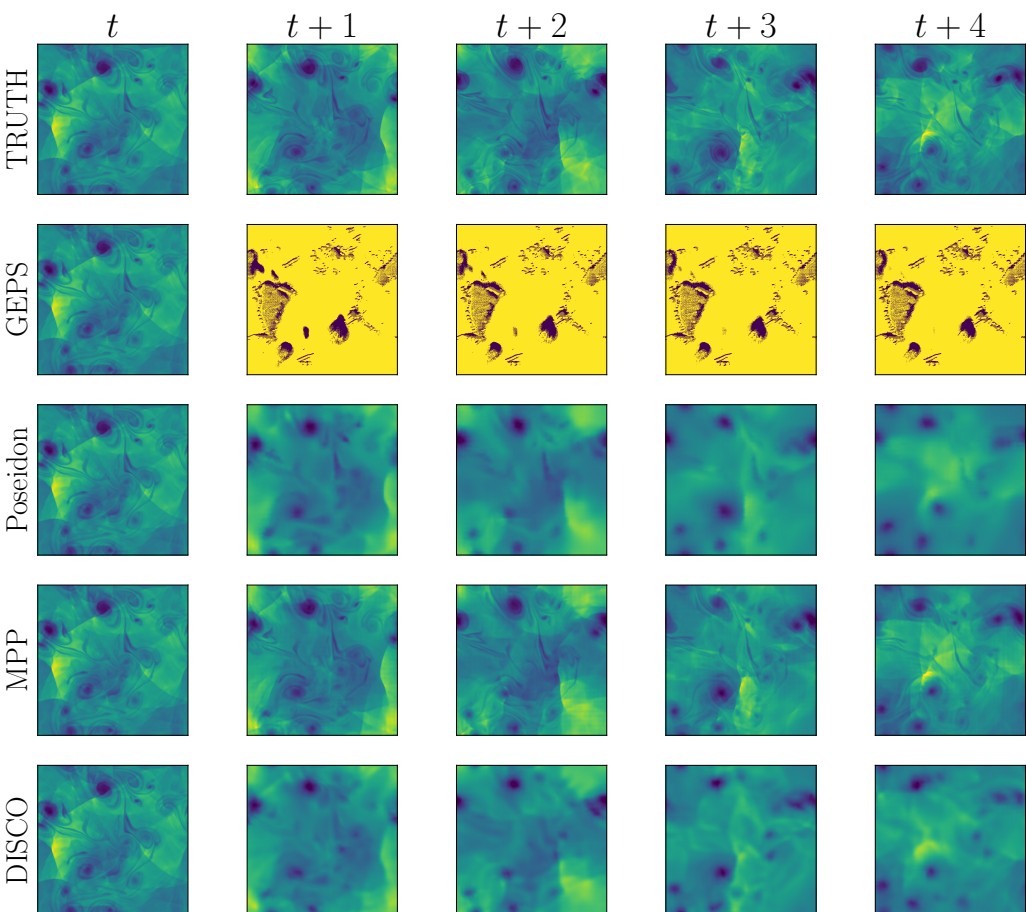

*Figure 9.* Example of rollout trajectory for the compressible Navier-Stokes dataset from PDEBench.

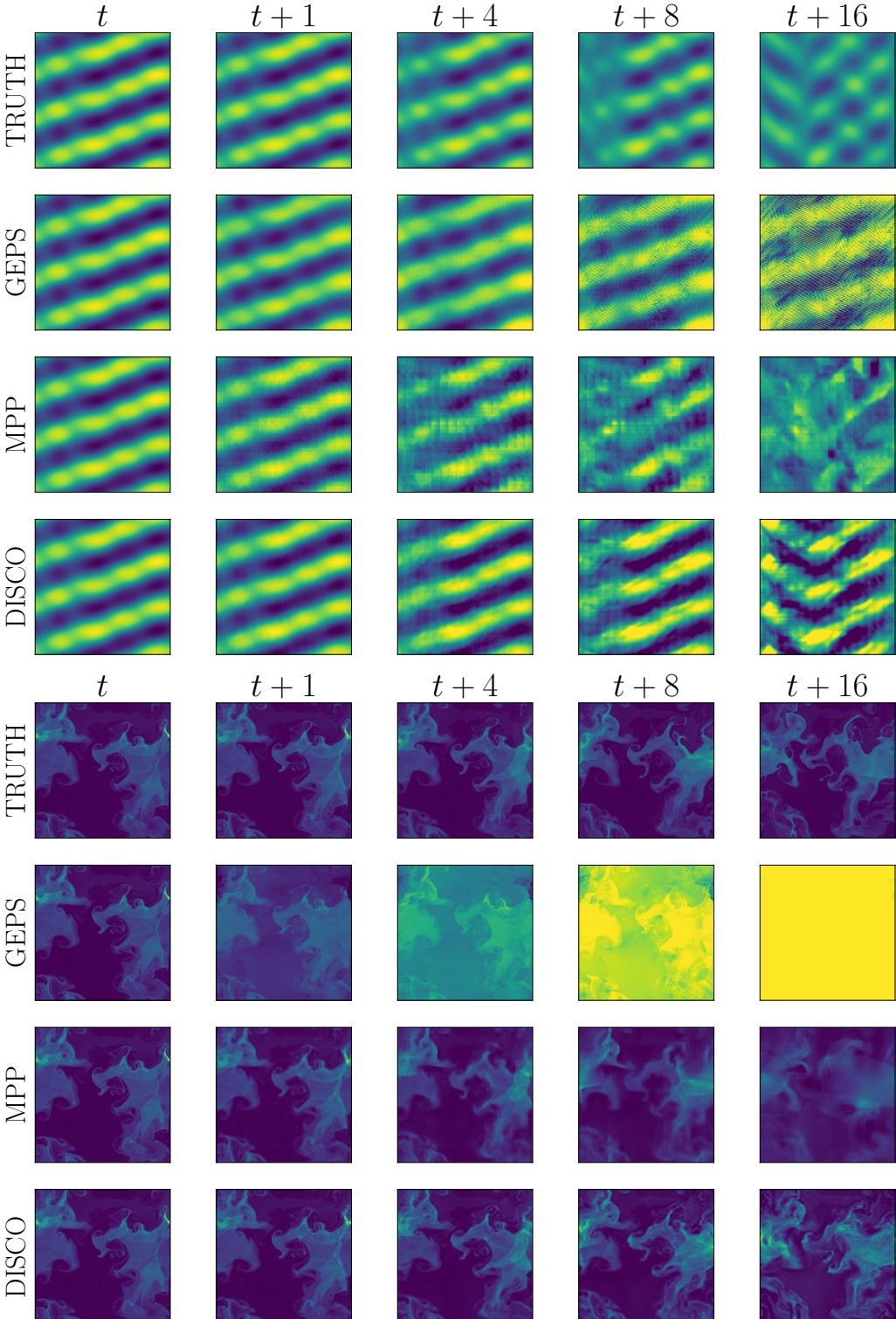

*Figure 10.* Examples of rollout trajectory for the active matter (top) and euler (bottom) datasets from the Well.

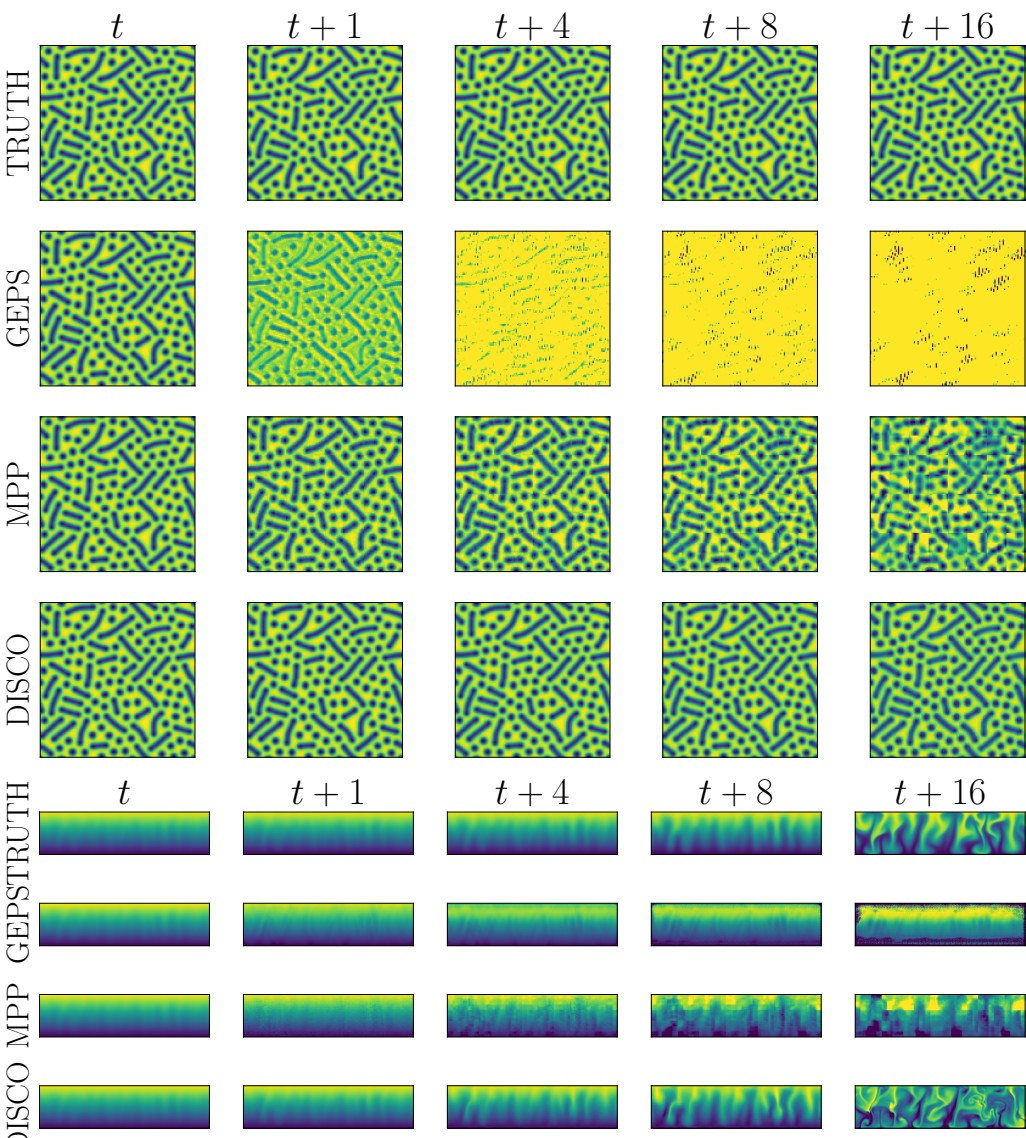

*Figure 11.* Examples of rollout trajectory for the Gray-Scott reaction-diffusion (top) and Rayleigh-Bénard (bottom) datasets from the Well.

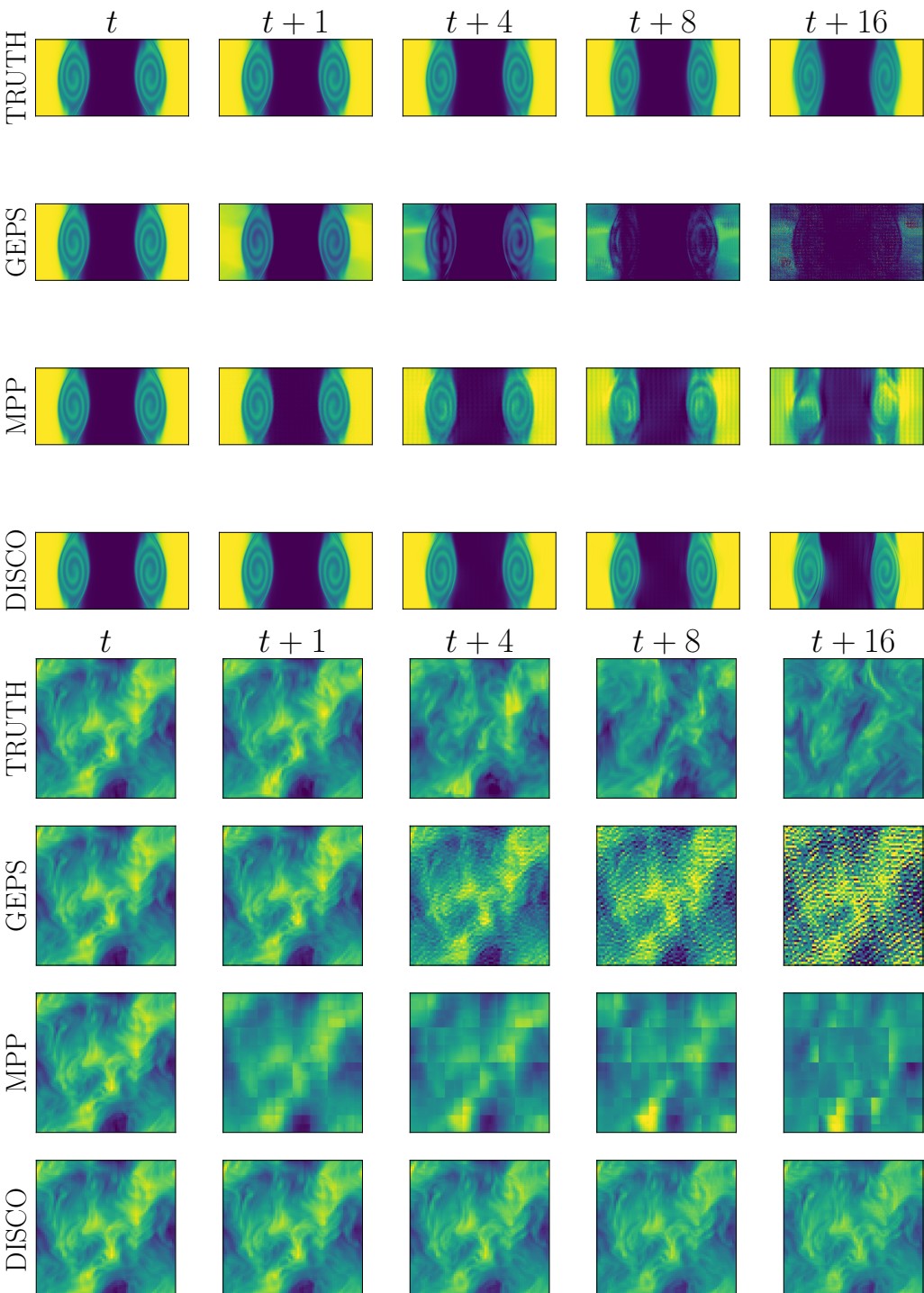

*Figure 12.* Examples of rollout trajectory for the shear flow (top) and MHD (bottom) datasets from the Well.

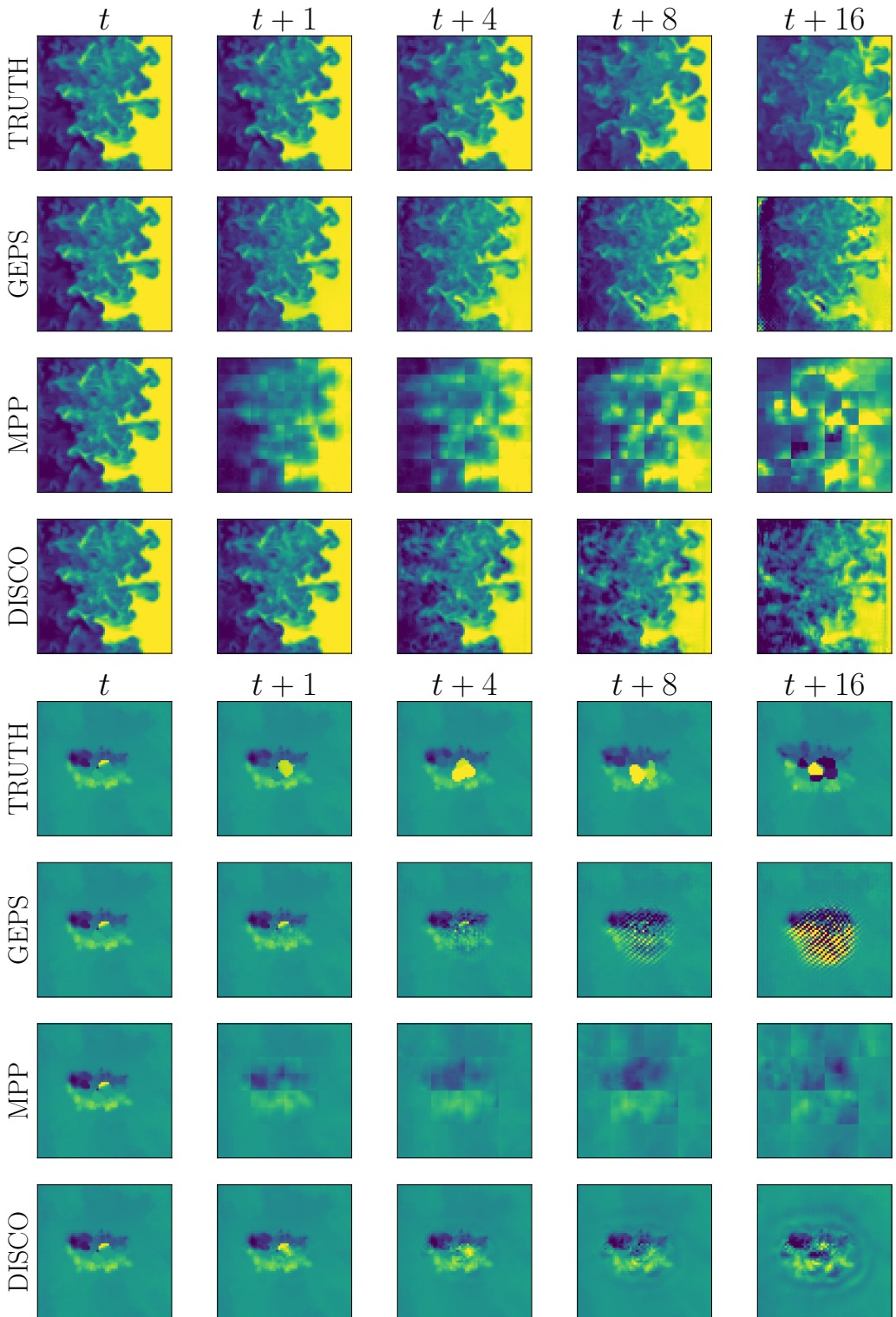

*Figure 13.* Examples of rollout trajectory (2D slices) for the Rayleigh-Taylor instability (top) and supernova explosion (bottom) datasets from the Well.

