# OpenReview forum: "DISCO: learning to DISCover an evolution Operator for multi-physics-agnostic prediction"
_ICML.cc/2025/Conference — ICML 2025 poster_

### Official Review · Reviewer_uxnz · 2025-03-10

**Overall Recommendation:** 4

**Summary:**

**Summary after rebuttal**

The authors resolved all of my concerns. I am raising to 4.

**End of Summary after rebuttal**

The paper introduces DISCO, a novel framework for multi-physics-agnostic prediction of dynamical systems governed by unknown temporal partial differential equations (PDEs). The key contribution is the use of a transformer-based hypernetwork to generate parameters for a smaller operator network, which then predicts the next state through time integration. This approach decouples the estimation of dynamics from state prediction, offering a more efficient and interpretable solution compared to traditional methods.

The only issues I have is the writing:

- There are some section mis-arrangements: eg, the network structure is introduced in the experimental results section, which should have been in the method. another example is the results section composites of both subsection and paragraph, which is chaotic for reading. The authors can consider only keeping the paragraphs, while moving the not so important sections into appendix.

- Undefined Terms/Symbols: Terms like d1 and d2 (dimensions of hypernet and operator net) are sparsely mentioned and not clearly defined at their first appearance, making it harder for readers to find. The authors should define all symbols at their first appearance and maintain consistency throughout.

- Clarity: The paper uses lots of technical jargon (e.g., "spatial translation equivariance") without sufficient explanation, which may hinder readability for a broader audience. I personally do not like this style of writing, and appreciate more clear writing and only keeping core concepts related to the contribution of this paper.

- Macro Formatting: The method name "DISCO" sometimes lacks a space before the following content. This could be resolved by adding a tilde (~) in LaTeX when using macros (e.g., "\macro~").

Besides the writing issue, the related work could be strengthened a bit. I will recommend a few citations to add in the below section of this review.

Overall, I give boarderline at this stage. The resolving of above problems will increase my score.

**Claims And Evidence:**

DISCO achieves state-of-the-art performance on next-step prediction across multiple physical systems in PDEBench.
Evidence: Table 2 shows DISCO outperforms MPP (McCabe et al., 2024) on most datasets with fewer epochs. For example, DISCO achieves NRMSE of 0.0027 on Burgers, compared to MPP's 0.0029.

DISCO generalizes well to unseen physics and initial conditions.
Evidence: Fine-tuning experiments on the Euler dataset show DISCO outperforms other models (Table 4). DISCO achieves NRMSE of 0.029, compared to 0.032 for MPP and 0.36 for GEPS.

**Essential References Not Discussed:**

The paper could benefit from discussing related work on using ML to learn stencils in PDE or CFD, as well as recent advances in graph neural networks (GNNs) for unstructured grids. Specifically:

a) Papers related to using ML to learn stencils in PDE or CFD; these works are highly similar to the current work, but for the meta-learning part:
- PDE-Net: Learning PDEs from Data
- Machine learning–accelerated computational fluid dynamics
- etc, authors can find much more by using citations of above


b) The authors mention MeshGraphNet for unstructured grids but could also discuss "Unet" for GNNs, similar to their using "Unet", instead of stacking CNN here:
- Multi-scale rotation-equivariant graph neural networks for unsteady Eulerian fluid dynamics
- Efficient Learning of Mesh-Based Physical Simulation with Bi-Stride Multi-Scale Graph Neural Network
- Learning Distributions of Complex Fluid Simulations with Diffusion Graph Networks
- etc,

c) Consider adding PointNet and PointNet++ ("Unet" ver of PointNet) as they can be applied to Lagrangian view simulations:
- PointNet: Deep Learning on Point Sets for 3D Classification and Segmentation
- PointNet++: Deep Hierarchical Feature Learning on Point Sets in a Metric Space
- etc,

**Experimental Designs Or Analyses:**

The paper evaluates DISCO on two collections of datasets: PDEBench (5 datasets) and The Well (9 datasets). These datasets cover a wide range of physical systems, including fluid dynamics, reaction-diffusion, and astrophysics.

 The paper compares DISCO against several baselines, including MPP (McCabe et al., 2024), Poseidon (Herde et al., 2024), and GEPS (Koupai et al., 2024). DISCO consistently outperforms these baselines in terms of accuracy and training efficiency.

**Methods And Evaluation Criteria:**

DISCO uses a transformer-based hypernetwork to generate parameters for a smaller operator network (U-Net). The operator network is integrated over time to predict the next state. The hypernetwork processes a context of successive states to infer the governing dynamics.

 Normalized Root Mean Square Error (NRMSE).
The paper evaluates performance on next-step prediction and multi-step rollouts across multiple datasets (PDEBench and The Well).

**Other Comments Or Suggestions:**

NA

**Other Strengths And Weaknesses:**

NA

**Questions For Authors:**

NA

**Relation To Broader Scientific Literature:**

The paper situates itself within the broader literature on neural PDE solvers and meta-learning for dynamical systems. It builds on recent work in transformer-based models for PDEs (e.g., MPP, Poseidon) and extends these approaches by introducing a hypernetwork to generate operator parameters.

**Theoretical Claims:**

The paper does not make strong theoretical claims but focuses on empirical performance. The theoretical justification for the use of hypernetworks and operator networks is grounded in classical numerical methods and finite difference schemes, which makes sense in my mind.

---

> ### Author Rebuttal · Authors · 2025-04-01
>
> We thank the reviewer for their thorough reading of our manuscript and for their valuable feedback on the writing and related works, which will help improve the paper.
>
> Since we cannot upload an updated manuscript for ICML, we will describe the changes we intend to make in response to your review below.
>
> Overall writing:
> - We will remove the "misarrangements" you mention. The "operator network architecture" and "hypernetwork architecture" paragraphs will be moved to subsection 3.2 to make this section, which introduces our model, more self-contained. As a result, subsection 4.1, "Generic architectures" will be removed. We agree that having only paragraphs in section 4 is beneficial for improving the readability of the paper. With small changes, section 4 will include the following paragraphs: "Multiple physics training", which briefly explains how training on multiple physics simultaneously is performed; "Datasets considered"; "Baselines"; "Next steps prediction performances"; "Information bottleneck and operator space"; "Fine-tuning on unseen Physics"; and "Model size ablation."
> - Undefined Terms/Symbols: The terms $d_1$ and $d_2$ will be defined more clearly and called “operator size” and “hypernetwork size” respectively throughout the paper. In particular, line 165 on the right column will be clarified: “where $\alpha\in \mathbb{R}^{d_2}$ are learnable parameters, while $\theta\in\mathbb{R}^{d_2}$ are parameters predicted by $\psi$, with $d_1$ and $d_2$ being the sizes of the operator network and hypernetwork respectively”.
> - Clarity: we think the term “translation equivariance”, often used in ML, is a well identified property, satisfied by many PDEs of interest (the right hand-side in Eq. (1) in the manuscript is translation equivariant). This property is reflected in the use of a translation equivariant U-Net as the operator network.  However, to clarify the meaning of this term we added a sentence when first introduced, line 076 “These methods often preserve key structural properties of physics, such as continuous-time evolution and translation equivariance [(Mallat, 1999)](https://www.sciencedirect.com/book/9780123743701/a-wavelet-tour-of-signal-processing) (i.e., a spatial translation of the initial condition results in the same translation of the solution, in the absence of boundary conditions), which transformers do not naturally inherit”.
> - Macro-formatting: these have been resolved.
>
> Related works. We thank the reviewer for the several relevant references!
> - The references to PDE-Net ([Long et al., 2021](https://proceedings.mlr.press/v80/long18a/long18a.pdf)) and [Kochkov, Smith et al. (2021)](https://www.pnas.org/doi/pdf/10.1073/pnas.2101784118) will be added line 99, as well as line 184 left column.
> - We will add the following sentence line 425 in the conclusion. “In particular, several papers, such as [Lino et al. (2022)](https://pubs.aip.org/aip/pof/article/34/8/087110/2847850) and [Cao et al. (2023)](https://proceedings.mlr.press/v202/cao23a.html), propose U-Net-like graph neural network architectures, which are natural candidates for the class of operator networks in DISCO. There are other promising directions ...”.
> To remain consistent and focused on the subject of our paper we will not include PointNet ([Qi, Su, et al. 2017](https://openaccess.thecvf.com/content_cvpr_2017/papers/Qi_PointNet_Deep_Learning_CVPR_2017_paper.pdf)) or PointNet++ ([Qi et al. 2017](https://proceedings.neurips.cc/paper_files/paper/2017/file/d8bf84be3800d12f74d8b05e9b89836f-Paper.pdf)), although we recognize them as important references.

---

> > ### Comment · Reviewer_uxnz · 2025-04-02
> >
> > Dear authors and other reviewers,
> >
> > I have read all the reviews and rebuttals. It seems the authors are respoding to them all very well. I also noticed sth interesting and hence would propose 2 new suggestions:
> >
> > - "Challenging to extend to different operator/PDEs"; I do have read this paper that can extrapolate to both different coefficients, or different combos of operators in a PDE. The authors should consider discussing this paper, in related work or future work.
> >
> >     - Towards a Foundation Model for Partial Differential Equations: Multi-Operator Learning and Extrapolation
> >
> > - "This method has better rollouts performance than MPP/MPP; MPP does not even report their rollout performance"; I do notice that MPP does not stand as the SOTA/best in this regard. The below two works both outperform MPP in rollouts, and they are also related (foundation models, also on 2D data). Though doing experiments is not needed, as you are doing meta learning. But I think they are related references worth discussing.
> >
> >     - DPOT: Auto-Regressive Denoising Operator Transformer for Large-Scale PDE Pre-Training
> >     - VICON: Vision In-Context Operator Networks for Multi-Physics Fluid Dynamics Prediction
> >
> > - Finally, the authors promised lots of revision on writing. Is it possible they include sth like "plan/list of revision" in their anonymous repo. This can help us judge if the writing quality should really improve, which is crucial for the community.
> >
> > Upon the above done, I will then be confident to raise score to 4.

---

> > > ### Author Response · Authors · 2025-04-09
> > >
> > > Dear reviewer uxnz,
> > >
> > > We thank you again for the valuable references and suggestions.
> > > - If you are referring to our response to reviewer Fp3h, as mentioned in our reply to them, our statement is made in the context of multi-physics agnostic prediction — the task addressed in our paper — where the model is not given the underlying equation, but only a few successive state observations. In the paper you mention ([Sun et al., 2025](https://arxiv.org/pdf/2404.12355)), the model is provided with the symbolic representation of the unseen operator as input (see Table. 2). In contrast, DISCO aims to discover an evolution operator from the data.
> > > - We thank you for mentioning these two references, which we will include our paper (see our list of revisions below). In short, they offer complementary methods that could potentially be combined with DISCO to enhance its performance
> > >   - DPOT’s noise injection, used to promote “generalization and scaling ability”, can be applied to the context fed to DISCO’s hypernetwork, and even to DISCO’s operator network. We expect further improvements in rollout accuracy. However, unlike DPOT, DISCO enforces a bottleneck to encourage the model to learn an actual evolution operator and provides a “space of operators” that can be interpreted (see Fig.3 in our paper). Finally, note that we haven’t seen any comparative rollout results on PDEBench with MPP on the DPOT paper.
> > >   - VICON. Instead of providing DISCO with a context made of frames from the same trajectory, we could also provide contexts composed of input-output pairs (where the input is the previous state and the output is the next state) to DISCO’s hypernetwork, similar to a standard in-context learning setting. More generally, contexts could include an arbitrary number of trajectories (see the Zebra model, [Serrano et al., 2024](https://arxiv.org/pdf/2410.03437)). Note that the VICON paper only reports comparisons with MPP on a single class of PDEs (compressible Navier-Stokes), and on 2D data downsampled to 128×128. In contrast, our paper reports results across all PDE classes MPP was trained on, and at the same resolution as MPP (up to 512×512 for certain datasets).
> > > - We acknowledge that clear and efficient writing benefits both the community and ourselves. [Here](https://hackmd.io/@anonymous-DISCO-icml/BkdsIzfCJl) is a list of the revisions we plan to make to our paper. We have aimed to provide a clear list without submitting a revised manuscript, as ICML does not permit this.

---

### Official Review · Reviewer_2f9m · 2025-03-11

**Overall Recommendation:** 4

**Summary:**

This paper proposes a novel method to obtain lightweight surrogate models from physics data. The idea is to use a hypernetwork transformer to learn the parameters of a smaller operator network, which is in charge of performing the time integration. This architecture decouples the learning of the dynamics from the state prediction, which is convenient for generalization to other domains and fine-tuning. The model is tested together with other baselines in two public benchmark datasets: PDEBench and The Well, showing state-of-the-art performance.

## Update after rebuttal
The authors addressed all my concerns satisfactorily, and I raised the score to 4.

**Claims And Evidence:**

All the claims of the paper are supported with validation results in benchmark cases. The method is novel and the results of the presented architecture clearly outperforms concurrent work baselines. I agree with the authors that the key feature of the method relies on the decoder-free structure (lines 290-294), which avoids the challenging (and sometimes, impossible) task of learning an inverse mapping.

**Essential References Not Discussed:**

It is mentioned in the paper that current literature is limited to known PDE equations or limiting adaptation to only the first layer (lines 138-145). However, one can found several examples in the literature about complete parametrized neural operators in the physics context. For example, a couple involving DeepONets and FNOs:

* [Lee, 2023] HyperDeepONet: learning operator with complex target function space using the limited resources via hypernetwork
* [Alesiani, 2022] HyperFNO: Improving the Generalization Behavior of Fourier Neural Operators

**Ethical Review Concerns:**

I have no ethical concerns.

**Experimental Designs Or Analyses:**

The experiments are based on two public benchmarks of challenging physics problems. They provide tests of multiple physics training, and finetuning on unseen physics. There is an additional visualization of the parameter space using Umap which shows that similar initial conditions induces similar latent parameters.

**Methods And Evaluation Criteria:**

The methods of this paper are clearly written and developed. The model uses a standard U-Net + Axial Vision Transformer architecture which is detailed in Appendix C. The output of the transformer is post-processed such that the U-Net parameters are conveniently normalized. The baseline networks are chosen to be recent state-of-the-art works such as GEPS or MPP. The evaluation criteria follows standard practices in physics-informed machine learning.

The rollout metrics in Table 3 and 6 are only integrated up to 16/32 timesteps. In my opinion, this is insufficient, as the datasets usually contain several hundreds of snapshots. It is a standard practise in the physics-informed deep learning field to do single-step supervision, but the test is always performed as a rollout from the initial conditions to several times the training time horizon. Single-step prediction error is informative, but useless when trying to evaluate real long-term predictions.

**Other Comments Or Suggestions:**

I have found some minor typos:
* Line 13: "unkown" might refer to "unknown".
* Line 233: "128=8.2^4" might refer to "128=8·2^4"
* Line 238: Space is missing in "1D,2D"
* Line 270-271: Spaces are missing in "1D,2D,3D".
* Lines 291-292: "we learn the to output the weights" might refer to "we learn to output the weights".
* Line 294: "cmopare" might refer to "compare"
* Table 3 caption: Spaces missing in "10,11,12,13".
* Lines 620-621: Spaces missing in "7,8,9" and "10,11,12,13".
* Line 760: Spaces missing in "(i),(ii),(iii)"
* Line 894: Spaces missing in "7,8,9".
* Figure 8: There is a number "20" on the third DISCO result image.

**Other Strengths And Weaknesses:**

I have no more comments.

**Questions For Authors:**

* Given that the operator network is relatively small, how does it compare to other NO methods, such as FNOs in terms of expresiveness? Have the authors tried to use a more expressive model?

* Table 3 and 6: Have the authors tried to rollout to further timesteps than t+16/t+32? It is very relevant to see how robust is the network and integration scheme over longer rollouts. A network can be overfit to match perfect single-step predictions but might not be robust over integration errors.

**Relation To Broader Scientific Literature:**

The literature is correctly discussed, appart from some missing references in regard to hyper-networks for PDEs (see next section).

**Theoretical Claims:**

There are no thoretical proofs in the paper. All the claims are validated experimentally as a pure data-driven procedure.

---

> ### Author Rebuttal · Authors · 2025-04-01
>
> We thank the reviewer for their positive feedback, in particular for acknowledging that “the methods of this paper are clearly written and developed” and that “all the claims of the paper are supported with validation results in benchmark cases”. We also thank the reviewer for their valuable suggestions and for pointing out several typos.
>
> Here are our answers to your questions:
> - We did try using a FNO ([Li et al., 2020](https://arxiv.org/abs/2010.08895)) for our operator network, as well as a U-Net with transformer blocks, but were unable to achieve better results while maintaining the same operator size (~$200$k parameters). To keep the FNO small, we had to reduce the hidden dimension, typically to $32$, and the number of Fourier modes in the spectral convolution layers, typically to $4$ per dimension, which limited the expressiveness of these architectures. A U-Net with transformer blocks faces similar challenges. For a fixed budget of ~$200$k parameters, allocating weights to the attention layers in the skip connections required reducing the hidden dimension, which in turn constrained the model’s expressiveness.
> - We chose not to include rollouts beyond $t+16$ / $t+32$ because neither DISCO nor any baseline models produced satisfactory results at those horizons. For $t+64$, most NRMSE values exceeded $1$, making comparisons difficult. That said, we agree with the reviewer that evaluating longer rollouts is important. Our results demonstrate that DISCO outperforms the main baselines GEPS ([Koupaï et al., 2024)](https://proceedings.neurips.cc/paper_files/paper/2024/file/82844e428d9163a9f94830dc03af4f9c-Paper-Conference.pdf)), Poseidon ([Herde et al., 2024](https://proceedings.neurips.cc/paper_files/paper/2024/file/84e1b1ec17bb11c57234e96433022a9a-Paper-Conference.pdf)), MPP ([McCabe et al. 2024](https://arxiv.org/abs/2310.02994)) in multi-step rollouts. Notably, the state-of-the-art model (MPP) does not report any metrics for multi-step predictions.\
> While tackling very long rollouts is not the goal of our work, we highlight in the conclusion that DISCO can naturally incorporate multi-step rollouts during training. Specifically, one can use DISCO’s hypernetwork (here, a transformer) to estimate the parameters meta-learned only once (one forward pass of hypernetwork), fix them and apply an integration scheme on a longer horizon than just $t+1$. Since the operator network is small, it will have a significantly lower memory usage than applying a large transformer such as MPP at each step in the future.
> - Regarding the last part of your second question, Fig. 11 (Appendix E) in our paper shows that even when trained on predicting the next step $t+1$, our model predicts the position of convection cells (mushroom-like spatial structures) quite decently after $t+8$, $t+16$. This is notable given that the future locations of these structures are highly sensitive to small perturbations in the state at $t$. This task is recognized as particularly challenging in the dataset’s paper (see “Sensitivity to initial conditions” in Appendix D in [Ohana et al., 2024](https://proceedings.neurips.cc/paper_files/paper/2024/file/4f9a5acd91ac76569f2fe291b1f4772b-Paper-Datasets_and_Benchmarks_Track.pdf)). Here are two additional examples of rollouts on the same dataset: [figure1](https://postimg.cc/64d79gYY), [figure2](https://postimg.cc/1V52GCzM). A model overfitting the next step prediction would have a hard time predicting correctly this behavior after $16$ steps.
>
> Here are some additional comments.
> - We appreciate that you have read some of the appendices (specifically, Appendix C as you write in your review), yet you also wrote “There is no supplementary material”. You should know that we have several appendices and that we also provided the code as a .zip file.
> - We will add the two relevant references you mentioned to our "Related works" section, but note the major differences: [Alesiani et al., 2022](https://ml4physicalsciences.github.io/2022/files/NeurIPS_ML4PS_2022_89.pdf) assumes the coefficients of the PDE are known and build a hypernetwork that take these coefficients as input, while we only have access to a short context of past states, making the prediction more challenging since our hypernetwork must infer the time evolution. [Lee et al., 2023](https://arxiv.org/abs/2312.15949) indeed uses meta-learning, but on fundamentally different objects and for different purposes. The authors employ a hypernetwork to predict the the parametrization of a future state, in the form of a network which takes the 2d coordinates and returns the value of the state at this location. In particular, such hypernetwork needs to be retrained every time the prediction horizon, the PDE coefficients, or the PDE class are changed. In essence, their hypernetwork predicts a state while on our task, our hypernetwork predicts an actual evolution operator.
> - The minor typos you pointed out will be corrected in the manuscript.

---

> > ### Comment · Reviewer_2f9m · 2025-04-03
> >
> > I thank the authors for the rebuttal. I only have one comment:
> >
> > * I still think that not being able to achieve decent long rollouts beyond 32 snapshots in a dataset composed of 100s/1000s snapshots is dissapointing, even though the other baseline are also not able to. However, I appreciate that the examples are very challenging and not easy to capture.
> >
> > Based on the rebuttal response, I have reconsidered my initial rating. I think the paper provides a substantial improvement over current methods, so I've raised my original rating to 4.

---

> > > ### Author Response · Authors · 2025-04-09
> > >
> > > Dear reviewer 2f9m,
> > >
> > > We greatly appreciate you raising your score. We assure the reviewer that we are working on achieving even longer rollouts, a task we, like the reviewer, consider important.
> > >
> > > Thanks

---

### Official Review · Reviewer_Fp3h · 2025-03-12

**Overall Recommendation:** 3

**Summary:**

The paper introduces DISCO, a novel framework for multi-physics-agnostic prediction that combines transformer-based hypernetworks with neural PDE solvers. The key innovation is a two-stage approach where a large transformer hypernetwork processes a context of sequential states to generate parameters for a smaller operator network, which then predicts future states through time integration. This architecture decouples dynamics estimation from state prediction, creating an information bottleneck that helps the model focus on essential dynamics rather than memorizing specific trajectories. The authors demonstrate that DISCO achieves state-of-the-art performance on benchmark datasets (PDEBench and The Well) while requiring significantly fewer training epochs than previous approaches.

**Claims And Evidence:**

The claims in the paper are well supported by substantial evidence from the extensive experimental evaluation. The authors provide detailed empirical results across two comprehensive datasets, with clear performance metrics for both next-step prediction and multi-step rollouts. Particularly compelling is the quantitative demonstration that DISCO achieves state-of-the-art performance with significantly fewer training epochs than transformer-based approaches. The paper also provides convincing evidence for generalization capabilities through visualization of the parameter space and fine-tuning experiments on unseen physics.

**Essential References Not Discussed:**

The authors mention neural PDE solvers but don't reference Jiang et al.'s "MeshfreeFlowNet: A Physics-Constrained Deep Continuous Space-Time Super-Resolution Framework".

**Experimental Designs Or Analyses:**

I examined the experimental designs and analyses in the paper and found them to be generally sound. The authors use appropriate datasets (PDEBench and The Well) that represent diverse physical systems, ensuring comprehensive evaluation.

**Methods And Evaluation Criteria:**

The proposed methods and evaluation criteria are highly appropriate for the multi-physics-agnostic prediction problem. The authors thoughtfully selected benchmark datasets (PDEBench and The Well) that span diverse physical phenomena across different spatial dimensions, providing a comprehensive testbed for their approach. Their evaluation metrics, particularly the normalized root mean square error (NRMSE) for both single-step and multi-step predictions, effectively capture model performance in practical settings. The comparison against multiple strong baselines (including transformers and meta-learning frameworks) strengthens the evaluation framework.

**Other Comments Or Suggestions:**

I did not identify any significant typos or grammatical errors in the paper

**Other Strengths And Weaknesses:**

Pros:

A particularly creative aspect is the architecture's information bottleneck design, which forces the model to learn generalizable physical laws rather than memorizing specific trajectories. This design choice shows originality in addressing overfitting challenges unique to physical system modeling.

Cons:

Though the authors mention potential applications, concrete real-world use cases would strengthen the paper's impact

**Questions For Authors:**

1. Given that DISCO uses a time integration method for prediction, how does the computational cost of inference compare to direct prediction methods like standard transformers?

2. The paper demonstrates impressive generalization to unseen physics when fine-tuning, but could you clarify whether DISCO can generalize to significantly different classes of PDEs (e.g., from diffusion-dominated to advection-dominated systems) without any fine-tuning?

3. The paper mentions that DISCO achieves better numerical accuracy on multi-step rollouts compared to baselines, but have you analyzed the model's ability to preserve important physical properties like conservation laws or symmetries during long rollouts

**Relation To Broader Scientific Literature:**

The key contributions of DISCO relate to several important research directions in the scientific literature. First, it builds upon recent advances in transformer-based models for physical system modeling (Yang et al., 2023; Liu et al., 2023; McCabe et al., 2024), but addresses limitations in their training efficiency and ability to preserve physical invariances. Second, DISCO connects to neural operator learning approaches (Li et al., 2020; Kovachki et al., 2023) while extending their capabilities to handle unknown and variable dynamics.

**Theoretical Claims:**

The paper is primarily empirical in nature and does not present formal mathematical proofs for theoretical claims. The authors establish conceptual connections between their approach and classical numerical methods for solving PDEs, particularly linking their operator network to finite difference schemes, but these are presented as motivational insights rather than rigorous theoretical proofs.

---

> ### Author Rebuttal · Authors · 2025-04-01
>
> We thank the reviewer for their positive comments on our work, particularly for acknowledging the “particularly creative aspect [of] the architecture's information bottleneck design” and noting that “the claims in the paper are well supported by substantial evidence”. We also appreciate their valuable suggestions and questions.
>
> Here are our answers to your questions
> 1. At inference time, on the Well datasets (see Table 1 in our paper), DISCO uses around two times fewer FLOPs than a transformer like MPP ([Mccabe et al., 2024](https://arxiv.org/abs/2310.02994)) to predict the next step $t+1$ (MPP: 4.0 GFLOPs, DISCO: 2.1 GFLOPs on a single context). Additionally, note that when predicting a large time horizon (e.g., $32$ as for this paper), one can use DISCO’s hypernetwork (here, a transformer) to estimate the parameters meta-learned only once (one forward pass of hypernetwork), fix them and apply an integration scheme on the long range-prediction $t+32$. On the contrary, MPP’s transformer, including its encoder and decoder, needs to be applied at each iteration. As a result, MPP requires significantly more FLOPs than DISCO: 128 GFLOPs vs. 37 GFLOPs for a horizon of $t+32$ from time step $t$.
> 2. No, without any additional modification, we do not expect DISCO to generalize to new classes of PDEs without fine-tuning (i.e., zero-shot generalization). This is a challenging task, and to our knowledge, no existing method designed for multi-physics agnostic prediction has demonstrated this capability. For comparison, both Poseidon ([Herde et al., 2024](https://proceedings.neurips.cc/paper_files/paper/2024/file/84e1b1ec17bb11c57234e96433022a9a-Paper-Conference.pdf)) and GEPS ([Koupaï et al., 2024](https://proceedings.neurips.cc/paper_files/paper/2024/file/82844e428d9163a9f94830dc03af4f9c-Paper-Conference.pdf)) require fine-tuning by design, even when applied to the same PDE. MPP ([Mccabe et al., 2024](https://arxiv.org/abs/2310.02994)) does not claim zero-shot generalization, and the only zero-shot result shown (Fig. 1) is not compelling.
> 3. This is a good suggestion. Given that we obtain better estimates compared to baselines such as MPP ([Mccabe et al., 2024](https://arxiv.org/abs/2310.02994)), our model by design better preserves the conservation laws, which is not surprising (see figure [here](https://postimg.cc/0KvhbXpV)). Since we use an autoregressive model that does not explicitly enforce conservation laws,
> these laws will gradually become less respected over time.
> Incorporating these conservation laws into our training, as done in PINNs ([Cai et al., 2021](https://arxiv.org/pdf/2105.09506)), would require knowing the specific conservation laws for each context. Such an assumption is incompatible with the multi-physics agnostic prediction task addressed in our paper. \
> More details on the [figure](https://postimg.cc/0KvhbXpV): it shows the conservation of mass $\||\mathrm{div} u\||$ and momentum  $\||\partial_t u - \nu\Delta u + u\cdot\nabla u + \nabla p\||$ over model rollouts from $t+1$ to $t+64$ averaged on $32$ trajectories from the validation set of the shear flow dataset (incompressible fluid, see Table 1. in the paper). The dashed line represents the conservation law as satisfied in the data. DISCO exhibits smaller deviations from the conservation laws compared to MPP ([Mccabe et al., 2024](https://arxiv.org/abs/2310.02994)).
>
> Here are comments on other points you raised.
> - The suggested reference to MeshfreeFlowNet ([Jiang, Esmaeilzadeh et al., 2020](https://arxiv.org/pdf/2005.01463)), which features a Rayleigh-Bénard convection dataset, very similar to one of the datasets used in our paper, will be added to the “Related Works” section (line 100).
> - “Concrete real world use-cases”: We completely agree with the reviewer on the importance of getting closer to real-world applications. One specific application we are currently exploring is the evolution of an unknown Physical system given limited observational data. In this context, DISCO provides an operator space (see Fig. 3 for a visualization), where all evolution operators encountered during training are mapped. When presented with data from an unseen physical system, one can identify the "closest" known operators and leverage them to refine and adapt the model, improving predictions on the new system.

---

### Decision · Program_Chairs · 2025-05-01

**Decision:**

Accept (poster)

**Comment:**

The paper introduces a two stage framework to disentangle dynamics emulation with state prediction through a hypernetwork transformer for the latter and a lightweight neural DE for the former.

Strengths identified:
* Novel idea and new contribution to the literature in NN based emulation of physical systems
* Comprehensive evaluation on well-known PDE benchmarks with multiple physics and demonstration of good performance to unseen physics

Weaknesses:
* Real-world applications is limited but possibly acceptable for this paper given its contributions are on the methodology side with sufficient evaluation on the benchmarks
* Some concerns on long rollout performance - authors acknowledged this as part of future work
* Writing concerns largely clarified in the rebuttal process.

All reviews concur in the contributions. The authors should carefully revise the writing in the paper as clarified in the rebuttal so that their paper is accessible and understandable to the community.